# Transfer-Controllable Policy for Model Protection in Deep Reinforcement Learning

## Abstract

Online deep reinforcement learning (DRL) suffers from sample inefficiency. This inefficiency challenges the training of effective policy models for complex tasks and demands substantial time and computing resources. As trained policy models can be transferred to other applications, protecting their intellectual property (IP) has become a pressing issue. To address this, we need to prevent unauthorized transfers for IP protection while maintaining transferability for future scalability. We propose the first Transfer-Controllable Reinforcement Learning (TCRL) framework. It has two key components: the Environment Randomization module generates unauthorized target-domain environments randomly, and the Transfer-Controllable module trains a policy model using source-domain and these unauthorized target-domain environments. This model resists transfer in unauthorized settings yet remains transferable in authorized ones. We validated the framework's effectiveness across various DRL environments and algorithms. The TCRL policy model is hard to transfer to similar unauthorized target-domain environments, but achieves source-domain-like performance in authorized ones. In the MuJoCo environment, our trained policy model attains $98.78\%$ of the source-domain performance in authorized target-domain environments, and only $50.38\%$ in unauthorized ones.

## 1 Introduction

Deep Reinforcement Learning (DRL) techniques have thrived in various AI fields, like video games Nie et al. (2024), board games Schrittwieser et al. (2020), and robot control Han et al. (2024); Haarnoja et al. (2024). However, significant expertise is needed to ensure their proper operation Miki et al. (2022). For example, by creating 8 different reward functions, including torque and joint speed costs, and adopting curriculum learning, researchers enabled legged robots to learn animal-like dynamic maneuvers Hwangbo et al. (2019). Also, training the AlphaGo policy model requires tens of millions of dollars and thousands of GPUs Silver et al. (2016). Given the high investment in time, resources, and expertise, protecting the intellectual property (IP) of policy models is crucial.

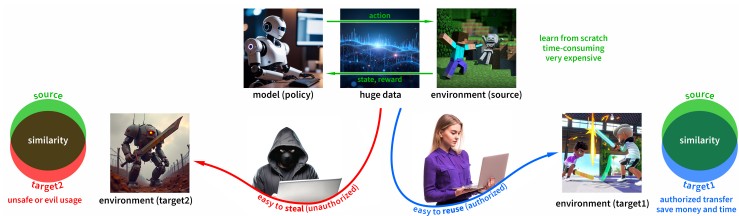

Figure 1: Training a policy model from scratch is time-consuming and costly. However, this model can be seamlessly transferred to similar scenarios, substantially reducing training expenses. To safeguard against theft by malicious actors, any transfer to unauthorized environments must be strictly prohibited. Simultaneously, to guarantee the model's scalability in future applications, its transferability within authorized environments should be maintained.

DRL policy models risk theft and unauthorized transfer. Their relatively small model size (Fig. 1) facilitates easy theft and quick transfer to similar domains. During policy training Silver et al. (2016), these models learn from observations and generate actions, storing valuable knowledge, making

them more vulnerable to theft than large datasets. Training a policy model from scratch is extremely time-consuming and costly, while using a pre-trained model can boost efficiency. Since policy models hold environment-related knowledge, it can be transferred to target domains via methods like learning from demonstrations, representation transfer, and inter-task mapping Yi et al. (2023). This transferability, however, also makes them prone to abuse.

Competitors may misuse obtained policy models by transferring them to similar scenarios, violating IP rights. For example, if a trained gameplay robot's policy model leaks, it could be used for illegal activities like poaching through transfer learning, as depicted in Fig. 1. However, completely banning model transfer across different environments would harm the open-source community and limit legitimate applications. With the growing use of DRL techniques, protecting policy model IP has become an urgent issue. To address this, we propose the first **Transfer-Controllable Reinforcement Learning (TCRL)** framework. This framework aims to balance model IP protection and usability in authorized environments. It has two main modules: the Environment Randomization module, which randomly generates unauthorized target-domain environments, and the Transfer-Controllable Training module. The latter optimizes data from the source and authorized domains and performs reverse optimization on unauthorized target-environment data. We also design a new policy-model objective to stabilize the training process. In our experiment, the transfer difficulty of all environments is set equally to ensure consistent experimental conditions. Our main contributions are as follows:

- We propose a new transfer-controllable task in DRL and validate its existence.

- We propose a preliminary TCRL framework to address this transfer-controllable task.

- Experimental results show policy models from our framework are controllably transferable: readily transferring to authorized target domains, yet struggling with unauthorized ones.

## 2 RELATED WORKS

**Policy Transfer in DRL.** Our work is the opposite of the goal of policy transfer. Policy transfer uses the knowledge learned on the source domain to help the policy training on the target domain Zhu et al. (2023). In policy distillation, the algorithms learn a student policy $\pi_{\theta_S}$ by minimizing the divergence of action distributions between the teacher policy $\pi_{\theta_T}$ and the student policy $\pi_{\theta_S}$ according to trajectories $\tau$. These studies can be further divided into two categories: teacher distillation Allen et al. (2021); Xu et al. (2019); Zhu et al. (2022) and student distillation D'Eramo et al. (2024); Schmitt et al. (2018). The difference between them is that $\tau$ is sampled from teacher policy: $\tau \sim \pi_{\theta_T}$ in teacher distillation and student policy: $\tau \sim \pi_{\theta_S}$ in student distillation. In policy reuse, the algorithms reuse a set of teacher policies by the means of $\pi$-reuse exploration strategy, which defines the trade-off among exploitation of the student policy, exploitation of the teacher policies, and exploration of random actions using the evaluation of the teacher policies' performance on the target domain. The typical research include Wu et al. (2024); Daoudi et al. (2024); Zhang et al. (2024); Gimelfarb et al. (2021); Tao et al. (2021); Yi et al. (2023); Tian et al. (2023).

**IP Protection in Deep Learning.** The IP protection in DRL is still in its infancy, whereas research on IP protection in Supervised Learning (SL) has made significant progress. In SL, the research can be divided into three main categories: digital watermarking, backdoor and fingerprint Xue et al. (2021); Fkirin et al. (2022). Digital watermarking involves embedding robust digital watermarks into SL models to protect the model IP rights Uchida et al. (2017). The side effect of digital watermarking that reduces the model prediction abilities is optimized from two aspects by backdoor Adi et al. (2018) and fingerprint Zhao et al. (2020). In DRL, some attack techniques are proposed to change the model output Behzadan & Munir (2017); Chen et al. (2021b), which shows that it is urgent to study countermeasures of IP infringement on DRL models Ilahi et al. (2021). Similar to the SL methods, some research in DRL also embeds watermarks into the target policy for ownership verification Behzadan & Hsu (2019); Chen et al. (2021a).

Different from the watermarking-based methods above, transfer-controllable learning restricts the generalization ability of the model on target domains while preserving its performance on source domains. The first approach of non-transfer learning was proposed in SL Wang et al. (2022). However, in DRL, the transfer-controllable learning problem has yet to be studied, and there are still many issues to be addressed in order to protect model IP. Compared to the large model size and stable training dataset in SL, the DRL model size is relatively small, and the dataset during training is unstable.

## 3 MOTIVATION

Unlike SL, policy model initialization in DRL is crucial Yi et al. (2023). Online DRL faces two major challenges: the exploration-exploitation dilemma and sparse rewards. The former requires balancing between using existing policies for rewards and exploring with stochastic policies; GoExplore addresses this by storing environmental states in an archive buffer Ecoffet et al. (2021). The latter occurs when agents need extended action sequences for non-zero rewards, which can be mitigated through immediate intrinsic rewards, as demonstrated with the 11 distinct rewards designed for a bipedal robot Duan et al. (2021). Addressing these challenges demands substantial resources in terms of funding, hardware, and training time.

However, if we have a better initial policy model before training, these difficulties can be alleviated Tirinzoni et al. (2019); Van Baar et al. (2019); Dennis et al. (2020); Abdolshah et al. (2021). A good initial model can perform correct actions, reducing the need for extensive exploration in the target environment to obtain sparse rewards Barreto et al. (2017); Wulfmeier et al. (2017); Riedmiller et al. (2018); Li et al. (2019); Guo et al. (2022).

Training a transfer-controllable policy model can resist transfer attacks and protect model IP rights. This raises two key questions: **(1) Is training such a model necessary? (2) What are the specific challenges in training transfer-controllable models in DRL compared to SL?** Given that a well-initialized policy model can reduce DRL training difficulty through transfer learning, answering these questions is significant.

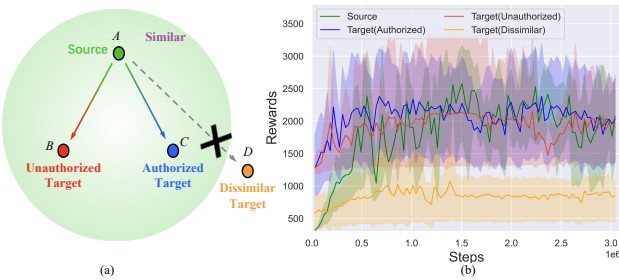

(a)                                        (b)

Figure 2: Preliminary Experiment. (a) We assume a green area exists where the source-domain policy model A can transfer to models B and C. Due to the policy overfitting in DRL, this green area is usually thought non-existent, meaning model A hardly transfers to model D. (b) The experimental results verify the existence of the green area. The Source, Target(Authorized), Target(Unauthorized), and Target(Dissimilar) curves correspond to policy models A, B, C, and D respectively.

**Necessity of Training Transfer-controllable Model.** To tackle Question (1), we first consider whether the policy model space contains similar regions. Such similarity is key as it enables a well-trained source-domain policy to transfer smoothly to certain target domains. In DRL, policies typically overfit to the source-domain environment, hindering their transfer to target domains. As shown in the dissimilar target domain in Fig. 2(a), transferring Policy A to Policy D is difficult. We assume there are similar target domains where the source-trained Policy A can quickly transfer to Policies B and C, as marked by the green area in Fig. 2(a).

To test our hypothesis, an experiment is conducted on the MuJoCo Hopper robot (Fig. 4(b)). The source domain featured Hopper parameters (torso, thigh, foot) of (0.05, 0.05, 0.06). Target domains, authorized, unauthorized, and dissimilar, have parameters (0.10, 0.05, 0.06), (0.05, 0.10, 0.06), and (0.2, 0.05, 0.06), respectively. Policy A, trained in the source domain, is transferred to these target domains. Results (Fig. 2(b)) show Policy A achieved 2000 reward in the source domain. In target domains with altered torso (Policy B) or thigh (Policy C) sizes, performance quickly reaches 2000. However, in the dissimilar target domain (Policy D) with a large torso change, Policy A's overfitting to the source domain hinders transfer. Results show training a transfer-controllable policy is essential. The source-domain trained policy has some transferability. We must prevent its transfer to unauthorized domains, while ensuring transfer to authorized ones for future scalability.

**Different Research Points on SL and DRL.** For Question (2), the research interests of transfer-controllable learning technology in DRL are distinct from those in SL. In SL, the main problem is how to overfit the source domain model to limit its generalization ability on the target do-

main Sadashivaiah et al.. The parameter space of the SL model is large, thus providing many directions for its optimization, making it easier to control the direction of overfitting while still ensuring the model's generalization on the target domain is limited. Furthermore, the datasets in SL are usually huge and stable, which makes the training process more stable and further reduces the difficulty of controlling the direction of overfitting. However, policy overfitting in DRL can limit the transfer of source domain policy models to certain target domain environments. However, in other target domain environments, the small model size of DRL models and the changing dataset distribution during its training process, bring more diverse problems in the DRL field. Therefore, it is necessary to conduct research on training transfer-controllable policy models.

# 4 METHODOLOGY

## 4.1 PRELIMINARY

In DRL, the agent learn from interaction with the environment, and the learning process is modeled with the Markov Decision Process (MDP) defined by a tuple $(S, A, P, r, \gamma)$. At each step $t$, the agent samples an action $a_t \in A$ from a policy distribution $\pi_\theta(a_t|s_t)$ where $s_t \in S$ is the observed state from the environment and $\theta$ is the policy model parameter. After passing the action $a_t$ into the environment, the environment transmits into the next state $s_{t+1}$ with the transition distribution $p(s_{t+1}|s_t, a_t) \in P$, and the agent receives a reward $r_t(s_t, a_t)$. Appendix A provides detailed explanations of each variable and foundational background on DRL.

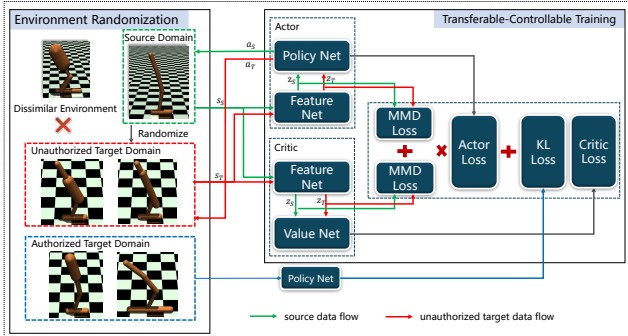

Figure 3: The main TCRL framework consists of Environment Randomization and Transfer-Controllable Training. The Environment Randomization module is used to randomly generate the unauthorized target domain environments and train some policy nets on authorized target domain environments, while the TCRL Training module trains the transfer-controllable policy model through a specific transfer-controllable loss function. The solid line represents the data interaction between the TCRL model and the environments, where the interaction targets of the red and green lines are the source domain and the unauthorized target domain environments, respectively.

## 4.2 TCRL FRAMEWORK

This paper introduces the TCRL framework (Fig. 3). The Environment Randomization module generates unauthorized target domains, uses user-provided authorized domains to train policies, and provides data for transfer-controllable training. The Transfer-Controllable Training module uses this output to train the transfer-controllable policy. Interactive source and unauthorized target data (Fig. 3) from source and generated target domains enable reverse transfer training, which limits transfer to unauthorized environments Also, the authorized-domain policy uses KL divergence for scalability.

### 4.2.1 ENVIRONMENT RANDOMIZATION MODULE

As depicted in the left part of Fig. 3, the Environment Randomization module generates unauthorized target-domain environments and concurrently creates several authorized policy models based on the user-provided authorized target domain. Initially, it randomly selects source-domain policy models to fine-tune the authorized policy models and collects offline datasets during the fine-tuning process. Subsequently, it randomly generates unauthorized target-domain environments according to specified rules, such as the two robot environments within the red box. Next, through model fine-tuning, their transferability is evaluated. Dissimilar target environments, marked with a red cross

---

**Algorithm 1** Environment Randomization Module

---

**Input:** environment parameters $\rho$ and Actor model set $P_{\text{model}} = \{\pi_\theta\}$ in source domain, parameter adjustment threshold $\delta$, user-provided authorized target-domain environments $E_{Auth}$

**Output:** environment parameter set $E_{\text{Unauth\_Target}} = \{\rho_i\}_{i=0}^N$, authorized target-domain Actor models $\pi^{Auth}$

1: Randomly select $\pi_\theta$ from $P_{\text{model}}$         ▷ Transfer authorized policy
2: Fine-tune $\pi_\theta$ on $E_{Auth}$ to get $\pi^{Auth}$
3: Initialize $i \leftarrow 0$, $E_{\text{Unauth\_Target}} \leftarrow \{\}$
4: **while** $i \leq N$ **do**             ▷ Parameter randomization
5:   Randomize parameters $\rho_i$ in the range $[\rho - \delta, \rho + \delta]$
6:   Construct $E_i$ through parameters $\rho_i$
7:   Randomly select $\pi_\theta$ from $P_{\text{model}}$       ▷ Model fine-tuning
8:   Fine-tune $\pi_\theta$ on $E_i$ to get reward $r_{\text{target}}$
9:   **if** Converge Time $t \leq T_{\text{threshold}}$ **then**
10:    $E_{\text{Unauth\_Target}} \leftarrow E_{\text{Unauth\_Target}} \bigcup \{\rho_i\}$, $i \leftarrow i + 1$
11:    Calculate the scaling factor $f_r = r_{\text{target}}/r_{\text{source}}$
12:   **end if**         ▷ Screening unauthorized target environments
13: **end while**

---

in Fig. 3, are excluded because they deviate significantly from the source-domain environment. The objective of this paper is to obtain a source-domain policy model that is difficult to transfer in previously unauthorized target-domain environments. Therefore, this module randomly generates target-domain environments and selects those that are easily transferable. We derived Theorem 1 to elucidate the existence of such unauthorized environments in the target domain.

**Theorem 1:** Let $\tau_S$ and $\tau_T$ represent all optimal trajectories in the source and target domains, respectively. For a given $\delta$, a state-action pair $(s_t, a_t, s_{t+1}) \in \tau_T$ is considered source-similar if there exists a state-action pair $(s'_t, a'_t, s'_{t+1}) \in \tau_S$ such that $|s_t - s'_t| < \delta$ and $|s_{t+1} - s'_{t+1}| < \delta$. Conversely, a state-action pair is considered target-specific if it is not source-similar. Then, an increase in the number of target-specific state-action pairs makes it more difficult to transfer to the target domain environment, and the $\mathcal{H}\Delta\mathcal{H}$ distance between the source and target domains satisfies

$$d_{\mathcal{H}\Delta\mathcal{H}}(\tilde{D}_S, \tilde{D}_T) \leq 2 \sup_{\eta \in \mathcal{H}_d} \left| \Pr_{\tilde{D}_S}[z : \eta(z) = 1] - \Pr_{\tilde{D}_T}[z : \eta(z) = 1] \right| \quad (1)$$

where $z$ denotes the feature of the state $s$, $\tilde{D}_S$ and $\tilde{D}_T$ represents the dataset on the source and target domain, respectively. The detailed proof for Theorem 1 is included in the Appendix B.

In detail, the algorithm process can be divided into four main phases: fine-tune authorized policy, parameter randomization, unauthorized model fine-tuning and screening environment, as shown in Algorithm 1. More details in the Appendix D.

### 4.2.2 Transfer-Controllable Training Module

The Transfer-Controllable Training module, illustrated in the right part of Fig. 3, is designed to train a transfer-controllable policy model in the source domain. This module interacts with the Environment Randomization module, as depicted in the middle of Fig. 3. During the model training process, the Actor model receives the source domain states $s_S$ and the target domain states $s_T$ from the environments in the Environment Randomization module at each step, and output the corresponding actions $a_S$ and $a_T$. Subsequently, the specific policy model objective is defined as

$$J_{\text{TCRL}}^{\theta_k, \mathcal{D}_{\text{Source}}, \mathcal{D}_{\text{Unauth\_Target}}, \mathcal{D}_{\text{Auth\_Target}}}(\theta) = J^{\theta_k, \mathcal{D}_{\text{Source}}}(\theta)$$
$$- \eta \cdot L_{\text{MMD}} \cdot J^{\theta_k, \mathcal{D}_{\text{Unauth\_Target}}}(\theta) + \lambda \cdot (\hat{D}_{KL}^{\mathcal{D}_{\text{Auth\_Target}}}(\pi_\theta(\cdot|s_t)||\pi^{Auth}(\cdot|s_t))) \quad (2)$$

where $J^{\theta_k, \mathcal{D}}(\theta)$ is defined in Eq. (6) and Eq. (7), $\theta_k$ denotes the Actor model parameters after $k$th training, $\mathcal{D}_{\text{Source}}$ and $\mathcal{D}_{\text{Unauth\_Target}}$ are data buffers, $\eta$ represents the learning rate of the reverse training, $\lambda$ represents the weighting factors for authorized scalability, and $\hat{D}_{KL}$ is the Kullback-Leibler divergence function. Meanwhile, the Feature Net in the Actor model outputs the intermediate features $z_s$ and $z_t$, and the maximum mean discrepancy (MMD) loss is computed as

$$L_{\text{MMD}} = \min \left( \alpha, \beta \cdot \left\| \sum_{i=1}^{n_1} \Phi(z_{s,i}) - \sum_{i=1}^{n_2} \Phi(z_{t,j}) \right\|_{\mathcal{H}}^2 \right) \quad (3)$$

---

**Algorithm 2** Transfer-Controllable Training Module

---

**Input:** environment parameter set $E_{\text{Unauth\_Target}} = \{\rho_i\}_{i=0}^{N}$ in target domain, maximum data buffer size $|\mathcal{D}|$
**Output:** transfer-controllable Actor model $\pi_\theta$
1: Initialize $k \leftarrow 0$, $\mathcal{D}_{\text{Source}} \leftarrow \{\}$, $\mathcal{D}_{\text{Unauth\_Target}} \leftarrow \{\}$          ▷ Algorithm prparation
2: Randomize the parameters of Actor $\pi_{\theta_k}$ and Critic $v_{\phi_k}$
3: Construct $E_{\text{Source}}$ and $\{E_k\}_{k=1}^{L}$ through $E_{\text{Unauth\_Target}}$ in each domain
4: **while** $k \leq N$ **do**          ▷ Data collection
5:      **while** $|\mathcal{D}_{\text{Source}}| + |\mathcal{D}_{\text{Unauth\_Target}}| \leq |\mathcal{D}|$ **do**
6:          Collect $\tau_S$ by running $\pi_{\theta_k}$ in source domain
7:          Collect $\tau_T$ by running $\pi_{\theta_k}$ in authorized target domain
8:          $\mathcal{D}_{\text{Source}} \leftarrow \mathcal{D}_{\text{Source}} \bigcup \{\tau_S\}$
9:          $\mathcal{D}_{\text{Unauth\_Target}} \leftarrow \mathcal{D}_{\text{Unauth\_Target}} \bigcup \{\tau_T\}$
10:      **end while**          ▷ Auxiliary variable calculation
11:      Compute $\hat{A}_t^{\text{Source}}$ and $\hat{A}_t^{\text{Unauth\_Target}}$ with Eq. (5)
12:      Compute $\hat{R}_t$ on $\mathcal{D}_{\text{Source}}$ and $\mathcal{D}_{\text{Unauth\_Target}}$ with $\hat{R}_t = \hat{A}_t + v_{\phi_k}(s_t)$
13:      **repeat**          ▷ Model parameter update
14:          Randomly choose $(s_t, a_t, z_t)$ from datasets $\mathcal{D}$
15:          Recompute $\pi_\theta(a_t|s_t)$ and $v_\phi(s_t)$
16:          Compute the MMD loss $L_{\text{MMD}}$ with Eq. (3)
17:          Update $\pi_{\theta_k}$ by maximizing $J_{\text{TCRL}}^{\theta_k}(\theta)$ through $\theta_{k+1} \leftarrow \theta_k + \nabla_\theta J_{\text{TCRL}}^{\theta_k}(\theta)$
18:          Update $v_{\phi_k}$ on $L_{\text{MSE}}(\phi)$ through $\phi_{k+1} \leftarrow \phi_k + \nabla_\phi L_{\text{MSE}}(\phi)$
19:      **until** $\mathcal{D}_{\text{source}}$ is empty
20:      $k \leftarrow k + 1$, $\mathcal{D}_{\text{Source}} \leftarrow \{\}$, $\mathcal{D}_{\text{Unauth\_Target}} \leftarrow \{\}$
21: **end while**

---

where $\Phi(\cdot)$ denotes the Gaussian kernel function, $\mathcal{H}$ indicates the Hilbert space, and $\alpha, \beta$ are the tunable hyperparameters.

The two equations above are essential for achieving anti-transfer training in the unauthorized target-domain environments while maintaining transferability in the authorized target-domain environments. In Eq. (2), the first term represents the model training in the source domain environment, while the second term indicates the reverse model training in the generated unauthorized target domain environments. There may be some similar samples on the source domain dataset $\mathcal{D}_{\text{Source}}$ and the target domain dataset $\mathcal{D}_{\text{Unauth\_Target}}$. This causes the gradients from the source domain $J^{\theta_k, \mathcal{D}_{\text{Source}}}(\theta)$ and the target domain $-J^{\theta_k, \mathcal{D}_{\text{Auth\_Target}}}(\theta)$ to be opposite, negatively impacting the model training on the source domain environment. To address this, factors $\eta$ and $L_{\text{MMD}}$ are introduced to adjust the strength of reverse training on the target domain environments, thus decreasing the negative impact. Additionally, the term $-L_{\text{MMD}}$ could increase the distribution distance between the source domain feature $z_s$ and the target domain feature $z_t$, making it easier to optimize in different directions on $\mathcal{D}_{\text{Source}}$ and $\mathcal{D}_{\text{Unauth\_Target}}$, thus reducing the difficulty of reverse training optimization. In addition, the third term $\hat{D}_{KL}^{\mathcal{D}_{\text{Auth\_Target}}}$ is used to ensure the transferability of the policy model in the authorized target-domain environment. Here, $\mathcal{D}_{\text{Auth\_Target}}$ is the fixed dataset obtained in the previous step, which is used to fine-tune the transfer-controllable policy model to the policy model in the authorized target domain. Since $\mathcal{D}_{\text{Auth\_Target}}$ is a fixed and small dataset, it has little impact on the overall training of the first two terms. From the domain adaptation theory, we derived Theorem 2 to illustrate the role of the MMD loss as follows

**Theorem 2:** Assume $p(s, a)$ is the joint distribution of state $s$ and action $a$. Given $\delta \in [0, 1]$, let a partition $\Omega \subseteq \mathbb{R}^n$ on the $\mathcal{H}$ space satisfies $P_{p(s,a)}(s \in \Omega) = \delta$, then
(1) there exists a partition $\Omega_{D_S}$ and $\Omega_{D_T}$ such that

$$d_{\mathcal{H}\Delta\mathcal{H}}(D_S, D_T) \geq 2\big|E_{s \sim D_S}[A(s) \neq A'(s)] - E_{s \sim D_T}[A(s) \neq A'(s)]\big| \tag{4}$$

(2) maximizing the MMD loss is equivalent to increasing the distance $d_{\mathcal{H}\Delta\mathcal{H}}$. The detailed proof for Theorem 2 is included in the Appendix B.

Concretely, the data processing flow includes four main phases: preparation, data collection, auxiliary variable calculation, and model parameter update, as shown in Algorithm 2. More details in the

Appendix D. The discounted reward $\hat{R}_t$ is calculated as

$$\hat{A}_t^{\mathcal{D}} = \sum_{\mathcal{D},l} (\gamma\lambda)^l (r_t + \gamma v_{\phi_k}(s_{t+l+1}) + \beta \cdot L_{\text{MMD}} \cdot (-\log(\pi(a_t|s_t)) \cdot \mathbf{1}_a + \epsilon) - v_{\phi_k}(s_{t+l})) \quad (5)$$

where $\mathcal{D}$ denotes the data buffer, $v_{\phi_k}$ denotes the Critic model, $\gamma$ and $\lambda$ are adjustment factors.

## 5 EXPERIMENTAL RESULT

### 5.1 EXPERIMENT SETUP

To verify the training effect of our framework and the performance of TCRL model obtained through training, we conducted experiments on different DRL algorithms and in different test environments.

**DRL Algorithms**, namely DQN Mnih et al. (2015) and PPO Schulman et al. (2017), are employed to comprehensively evaluate the performance of these algorithms under various conditions, aiming to uncover their respective advantages and limitations in solving the targeted problems.

**Test Environments.** The main body of the text primarily presents the experimental results of our framework in the Maze Environment and the MuJoCo Environment Todorov et al. (2012). The configuration examples of these environments are illustrated in Fig. 4. In the experiments conducted in the Maze Environment, we test the effectiveness of the DQN algorithm within our framework. Meanwhile, in the experiments carry out in the MuJoCo Environment, we examine the performance of the PPO algorithm within the same framework. Additional experimental results under various settings are available in Appendix E.

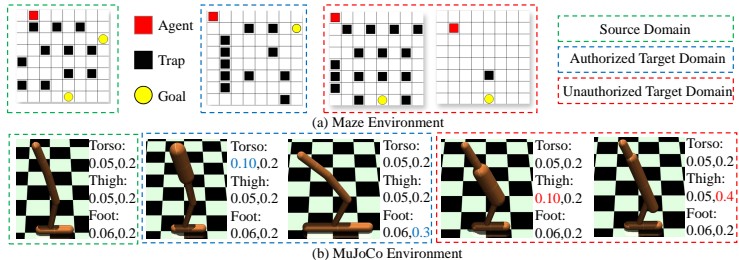

(a) Maze Environment

(b) MuJoCo Environment

Figure 4: Overview of Experiment Setup. (a) Maze Environment. It consists of the Agent, Trap, and Goal. In the source domain, there are two Goals, one on the right and one at the bottom. The authorized target domain has a single Goal on the right, while the unauthorized target domain has only one Goal at the bottom. In independent experiments, the positions of the Agent, Trap, and Goal vary; (b) MuJoCo Environment. It encompasses MuJoCo robots with diverse configurations. In the authorized target domain, users set the configurations based on the subsequent scalability requirements of the model. In contrast, configurations in the unauthorized target domain are randomly generated by the Environment Randomization module. In this example, users primarily specify the Torso and Foot configurations of the Hopper robot, while the Thigh configuration is generated by the Environment Randomization module.

### 5.2 PERFORMANCE OF DQN ON MAZE ENVIRONMENT

In this experiment, Agent receives a final reward of 60 upon reaching Goal and -10 if it enters Trap by mistake. A single experiment terminates when Agent reaches the Goal or the environment runs for more than 200 steps. As shown in Fig. 5(a), both the original DQN (blue curve) and TCRL_DQN (orange curve) can achieve a reward value of around 50 during training in the source domain, indicating that the trained Agents can complete the tasks. This implies that our method has little impact on the performance of the source-domain policy model during training.

Fig. 5(b) reveals that in the policy model transfer experiment, the transfer-controllable policy model trained by the TCRL framework (orange curve) can complete the task in the authorized environment but struggles to do so in the unauthorized environment. Here, TCRL_Trans_Unauth_1 (green curve) and TCRL_Trans_Unauth_2 (red curve) correspond to the two experimental settings in the red box of Fig. 4(a) respectively. The red curve shows that even when the number of Traps is significantly reduced, the transfer-controllable policy model still fails to complete the task. This demonstrates

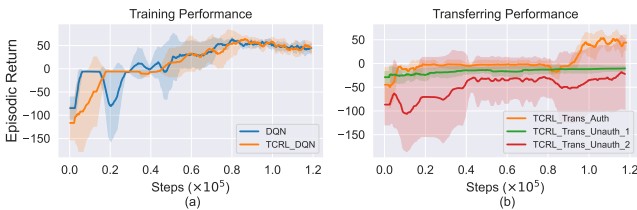

Figure 5: Experiment of DQN on Maze Environment.

that the policy model obtained through our training exhibits strong reverse transfer ability in the unauthorized environment.

### 5.3 TRAINING PERFORMANCE OF PPO ON MUJOCO ENVIRONMENT

In this experiment, we aim to verify the effect of the TCRL algorithm on the training performance of the baseline algorithm. To do so, we used 32 copies of the same source domain environment to train the benchmark PPO algorithm in parallel, and employ the same 32 source domain environment copies, as well as 32 unauthorized target domain environments, to train the TCRL algorithm in parallel. The convergence of the reward curve is used as the evaluation criteria.

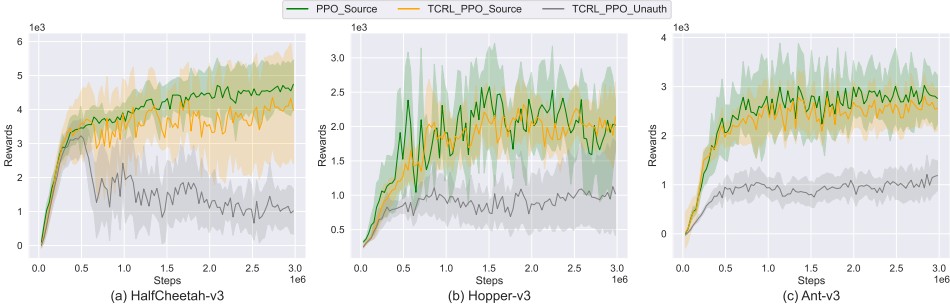

Figure 6: Training performance of the baseline PPO algorithm and our TCRL algorithm. The blue $PPO\_Source$ and orange $TCRL\_PPO\_Source$ curves denote the performance of PPO and TCRL on the source domain, while the green $TCRL\_PPO\_Unauth$ curves indicate the performance of TCRL on the unathorized target domains.

The experimental results show that during training, TCRL can achieve a performance similar to PPO in the source domain. Meanwhile, it can significantly degrade the performance of the policy model in the unauthorized target domain. In Fig. 6, for the HalfCheetah, Hopper, and Ant tasks, the TCRL_PPO_Source curve converges to an average reward value close to that of the PPO_Source curve, though with a slightly larger variance. This indicates that our method may slightly increase the training difficulty of the algorithm, but has minimal impact on the final training outcome, as both can yield effective policy models. On the other hand, the TCRL_PPO_Unath curve is limited to a very low value in the unauthorized target domain environment. This demonstrates that our method restricts the policy model's performance in such environments, laying the groundwork for subsequent transfer experiments.

### 5.4 TRANSFERRING PERFORMANCE OF PPO ON MUJOCO ENVIRONMENT

In this experiment, we aim to verify the effectiveness of the obtained transfer-controllable policy model in preventing the transfer of the source domain to the unauthorized target domain. To do so, we use the trained PPO and TCRL policy models to transfer on 8 authorized and 32 unauthorized target domain environments, respectively. Additionally, a random initialized policy model was trained under the same target domain environment as a benchmark. Subsequently, the policy models were tested on 8 authorized and 8 unauthorized target domain environments to verify the average transfer-controllable ability during training process. The convergence of the reward curve was then used as the evaluation criterion for transfer performance.

Based on the experimental results, it is evident that the TCRL policy model can effectively impede the transferability of the source-domain policy model to the unauthorized target domain. As depicted in Fig. 7, the reward values achieved by the TCRL_PPO_Trans_Unauth curve are substantially lower

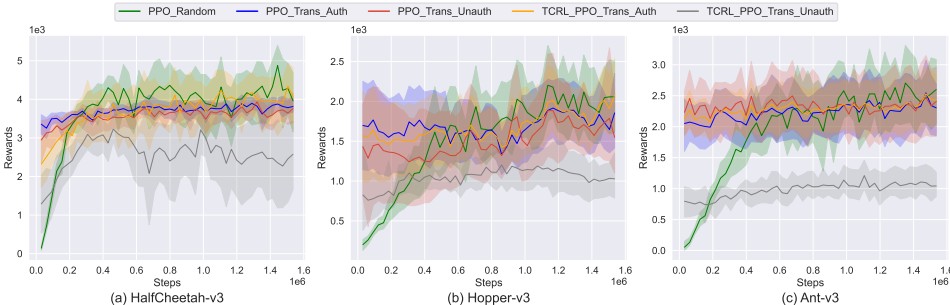

Figure 7: Comparing the transfer performance between the PPO and TCRL policy models. The $PPO\_Trans\_Auth$ curve denotes the PPO model and the $TCRL\_PPO\_Trans\_Auth$ curve denotes the TCRL model on authorized target domain, while the $PPO\_Trans\_Unauth$ curve denotes the PPO model and the $TCRL\_PPO\_Trans\_Unauth$ curve denotes the TCRL model on unauthorized target domain. The green $PPO\_Random$ curve, trained with a random initialized model, serves as the baseline.

Table 1: Transferring performance on MuJoCo environment. $PPO\_Random$ denotes the average rewards in authorized and unauthorized target-domain environments. $TCRL\_Trans\_Auth$ and $TCRL\_Trans\_Unauth$ represent the average rewards of TCRL policy model transfer to authorized and unauthorized target-domain environments, respectively.

|  | HalfCheetah-v3 | Hopper-v3 | Ant-v3 | Mean |
|---|---|---|---|---|
| PPO_Random | 4123.66 | 2057.22 | 2598.95 | - |
| TCRL_Trans_Auth | 4207.33 | 2075.46 | 2427.83 | - |
| Ratio | 102.03% | 100.89% | 93.42% | 98.78% |
| TCRL_Trans_Unauth | 2516.65 | 1028.16 | 1043.02 | - |
| Ratio | 61.03% | 49.98% | 40.13% | 50.38% |

than those of the PPO_Trans_Unauth curve. This implies that the TCRL policy model encounters significant difficulties in migrating to the unauthorized target-domain environment, thereby demonstrating a robust anti-transfer capacity. The PPO_Random curve represents the average reward values obtained through training from the initial state in each task environment. The convergence values of the PPO_Trans_Unauth curve are comparable to those of the PPO_Random curve. This indicates that the original PPO algorithm is essentially incapable of preventing the source-domain policy model from transferring to the unauthorized target-domain environment.

Meanwhile, Fig. 7 clearly demonstrates that TCRL policy model preserves its transferability within the authorized target domain, thereby providing an avenue for subsequent model expansion. The convergence values of the TCRL_PPO_Trans_Auth curve exhibit minimal divergence from those of the PPO_Trans_Auth curve and closely approximate the reward values of the PPO_Random. This observation implies that our proposed policy model retains a high-level of transferability.

As indicated in Table 1, the TCRL policy model derived from our training regimen not only sustains a transfer performance of 98.78% in the authorized target domain but also effectively restricts the transfer performance of the policy model to the unauthorized target domain to 50.38%.

## 6  CONCLUSION AND LIMITATION

In this paper, we have introduced a new task of training transfer-controllable policies in DRL and presented an original framework to address this task. Firstly, we have examined the necessity of transfer-controllable learning in DRL and identified potential challenges that may arise. Subsequently, we proposed the TCRL framework for transfer-controllable training and theoretically demonstrated its feasibility. Moreover, we applied this framework to obtain a transfer-controllable policy model and empirically validated its efficacy in safeguarding against transfer attacks on the policy model. However, the TCRL framework's major limitation is that it consumes approximately twice the computational resources of conventional DRL training, mainly because of the high computational cost of stochastically generating suitable unauthorized target domain.

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

## A  USE OF LLMs

This paper did not use any LLMs during the research, writing, and other related processes.

## B  FOUNDATIONAL BACKGROUND ON DRL

### B.1  DQN AND PPO OBJECTIVES AND CRITIC LOSS

In this paper, both the deep $Q$-network (DQN) and the proximal policy optimization (PPO) Schulman et al. (2017) algorithms are used to train the policy model. The main objectives of DQN and PPO are:

$$J_{\text{DQN}}^{\theta_k, \mathcal{D}_k}(\theta) = \hat{\mathbb{E}}_{t, \mathcal{D}_k} \left\{ \left[ Q(s_t, a_t; \theta) - r_t(s_t, a_t) - \gamma \max_{a'} Q(s_{t+1}, a'; \theta_k) \right]^2 \right\}, \tag{6}$$

and

$$J_{\text{PPO}}^{\theta_k, \mathcal{D}_k}(\theta) = \hat{\mathbb{E}}_{t, \mathcal{D}_k} \left\{ \min \left[ \frac{\pi_\theta(a_t|s_t)}{\pi_{\theta_k}(a_t|s_t)} \hat{A}_t, \, \text{clip} \left( \frac{\pi_\theta(a_t|s_t)}{\pi_{\theta_k}(a_t|s_t)}, 1 - \epsilon, 1 + \epsilon \right) \hat{A}_t \right] \right\}, \tag{7}$$

where $Q$ represents the action-value function, $\theta_k$ indicates the network parameters of the old policy model at the $k$th training epoch, $\mathcal{D}_k$ denotes the data buffer at the $k$th training epoch, $\epsilon$ is a hyperparameter, and $\hat{A}_t$ indicates the advantage estimates.

The Critic loss $L_{\text{MSE}}$ is defined as

$$L_{\text{MSE}}(\phi) = \hat{\mathbb{E}}_{t, \mathcal{D}_k}[(V_\phi(s_t) - \hat{R}_t)^2] \tag{8}$$

where $\mathcal{D}_k$ indicates the data buffer of the chosen $(s_t, a_t, z_t)$ pairs from $\mathcal{D}_{\text{Source}}$ and $\mathcal{D}_{\text{Unauth\_Target}}$.

### B.2  SYMBOL DEFINITIONS

The symbols used in this paper and their corresponding meanings are shown in Table below.

## C  THEORY PROOFS

**Theorem 1:** Let $\tau_{D_S}$ and $\tau_{D_T}$ represent all optimal trajectories in the source and target domains, respectively. For a given $\delta$, a state-action pair $(s_t, a_t, s_{t+1}) \in \tau_{D_T}$ is considered source-similar if there exists a state-action pair $(s'_t, a'_t, s'_{t+1}) \in \tau_{D_S}$ such that $|s_t - s'_t| < \delta$ and $|s_{t+1} - s'_{t+1}| < \delta$. Conversely, a state-action pair is considered target-specific if it is not source-similar. Then, an increase in the number of target-specific state-action pairs makes it more difficult to transfer to the target domain environment, and the $\mathcal{H}\Delta\mathcal{H}$ distance between the source and target domains satisfies

$$d_{\mathcal{H}\Delta\mathcal{H}}(\tilde{D}_S, \tilde{D}_T) \leq 2 \sup_{\eta \in \mathcal{H}_d} |\text{Pr}_{\tilde{D}_S}[z : \eta(z) = 1] - \text{Pr}_{\tilde{D}_T}[z : \eta(z) = 1]| \tag{9}$$

Table 2: Symbol Definitions

| Symbol | Notation |
|---|---|
| $t$ | The current time step |
| $k$ | The $k$th training epoch |
| $S \in \mathbb{R}^m$ | The state space |
| $A \in \mathbb{R}^n$ | The action space |
| $\mathcal{H}$ | The Hilbert space |
| $P : S \times A \times S \to \mathbb{R}^+$ | The state transition distribution |
| $r : S \times A \to \mathbb{R}$ | The reward function |
| $\gamma \in [0, 1]$ | The discounted factor |
| $s_t \in S$ | The observed state from the environment at time step $t$ |
| $a_t \in A$ | The agent action at time step $t$ |
| $\Omega \subseteq \mathbb{R}^n$ | The partition on the $\mathcal{H}$ space |
| $z_t, f_t$ | The feature of the state $s_t$ |
| $p(s_{t+1}|s_t, a_t) \in P$ | The transition distribution at time step $t$ |
| $r_t(s_t, a_t)$ | The environment reward at time step $t$ |
| $(s_t, a_t, s_{t+1})$ | The state-action pair at time step $t$ |
| $\theta$ | The policy model parameter |
| $\phi$ | The value network parameter |
| $\theta_k$ | The policy model parameter at $k$th training epoch |
| $\phi_k$ | The value network parameter at $k$th training epoch |
| $\pi_\theta(a_t|s_t)$ | The policy distribution at time step $t$ with model parameter $\theta$ |
| $r_t(\theta)$ | The policy probability ratio with model parameter $\theta$ |
| $\mathcal{D}_k$ | The data buffer at the $k$th training epoch |
| $\hat{A}_t$ | The advantage estimates at time step $t$ |
| $J^{\theta_k, \mathcal{D}_k}(\theta)$ | The main optimized objective of the PPO and the DQN algorithm |
| $J_{\text{TCRL}}^{\theta_k, \mathcal{D}_{\text{Source}}, \mathcal{D}_{\text{Unauth\_Target}}}(\theta)$ | The specific policy model optimized objective of the TCRL algorithm |
| $L_{\text{MMD}}$ | The maximum mean discrepancy loss |
| $\tau_S, \tau_T$ | The optimal trajectories in the source and target domains, respectively |
| $\tilde{D}_S, \tilde{D}_T$ | The dataset on the source and target domain, respectively |
| $\mathcal{D}_{\text{Source}}$ | The data buffer on the source domain |
| $\mathcal{D}_{\text{Unauth\_Target}}$ | The data buffer on the unauthorize target domain |
| $\Omega_{D_S}, \Omega_{D_T}$ | The partition on source and target domain dataset, respectively |
| $\rho$ | The environment parameters |
| $\epsilon, \delta$ | The hyperparameters representing small values |
| $\alpha, \beta$ | The tunable hyperparameters |
| $min(\cdot)$ | The minimize function |
| $clip(\cdot)$ | The clip function |
| $\mathbb{E}(\cdot)$ | The expected function |
| $U(\cdot)$ | The uniform distribution |
| $\Phi(\cdot)$ | The Gaussian kernel function |
| $\Pr(\cdot)$ | The probability function |

where $z$ denotes the feature of the state $s$, $\tilde{D}_S$ and $\tilde{D}_T$ represents the dataset on the source and target domain, respectively.

**Proof:** Firstly, the RL transfer problem needs to be transformed into an SL optimization problem.

Assume that a trajectory $\tau$ is randomly selected from the set of target domain trajectories $\tau_{D_T}$. If any state-action pair $(s_t, a_t, s_{t+1}) \in \tau$ is source-similar, the optimal actions $a_t$ and $a'_t$ satisfies that $|a_t - a'_t| < \Delta$ as the state assumption conditions that $|s_t - s'_t| < \delta$ and $|s_{t+1} - s'_{t+1}| < \delta$. That is, if the state-action pairs on the target domain are all source-similar, then these optimal actions $a_t$ and $a'_t$ can be divided into different categories.

Moreover, if a state-action pair $(s_t, a_t, s_{t+1}) \in \tau$ is target-specific, suppose that $|s_t - s'_t| < \delta$ and $|s_{t+1} - s'_{t+1}| < 2\delta$, then there exists a state-action pair $(s''_t, a''_t, s''_{t+1})$ satisifies that $|s''_t - s'_t| < \delta$ and $|s''_{t+1} - s'_{t+1}| < 2\delta$. Then, the optimal actions satifies that $|a''_t - a'_t| < \Delta$ and $|a''_t - a_t| < \Delta$, and it means that $|a_t - a'_t| < 2\Delta$. Furthermore, if there are few target-specific points in the target domain, these optimal actions $a_t$ and $a'_t$ can be divided into different categories through the auxiliary action $a''_t$.

For the optimal trajectories $\tau_{D_S}(s_i) = [s_0, a_0^{opt}, ..., a_{i-1}^{opt}, s_i]$, given the Markov property, optimizing $\tau_{D_S}$ in the source domain is equivalent to the existence of a classifier from state $s_i$ to action $a_i$ as $A_{D_S}(s_i) = a_i^{opt}$. Similarly, for the target domain, the optimal trajectories $\tau_{D_T}$ is equivalent to the optimal classifier $A_{D_T}(s_i) = a_i^{opt}$ from state $s_i$ to action $a_i$.

Then, we derive the $\mathcal{H}\Delta\mathcal{H}$ distance between the source and target domain.

Let the action space be $\mathcal{A}$. Since the action category space $\mathcal{A}_{D_S}$ and $\mathcal{A}_{D_T}$ are subsets of the action space $\mathcal{A}$, and both the source domain classifier $A_{D_S}$ and the target domain classifier $A_{D_T}$ satisfy

$$A_{D_S} \subseteq \mathcal{A}_{D_S} = \mathcal{A} \quad \text{and} \quad A_{D_T} \subseteq \mathcal{A}_{D_T} = \mathcal{A} \tag{10}$$

the attribute of $\mathcal{A}_{D_S}$ and $\mathcal{A}_{D_T}$ is the same. Considering the network architecture of the policy model $\pi_\theta$, assume the feature extraction function $f_{D_S}$ of the Feature Net statifies $f_{D_S}(s_i) = z_i \in \mathcal{Z} \subseteq \mathbb{R}^m$. As all classification problems can be transformed into binary classification Goodfellow et al. (2016), only the binary categories will be taken into consideration as $h : \mathcal{Z} \to \{0, 1\}$. Based on the domain adaptation theory Ben-David et al. (2010), for the classifier $A = h \circ f$, the error of the given classifier $h(z)$ on the target domain $D_T$ is

$$\epsilon_{D_T}(h) < \epsilon_{D_S}(h) + \frac{1}{2} d_{\mathcal{H}\Delta\mathcal{H}}(\tilde{D}_S, \tilde{D}_T) + \lambda \tag{11}$$

where $\epsilon_{D_T}(h)$ and $\epsilon_{D_S}(h)$ denote the error of the given classifier $h(z)$ on the source and target domain, respectively. The variable $d_{\mathcal{H}\Delta\mathcal{H}}$ represents the generalized distance between data buffer $\tilde{D}_S$ and $\tilde{D}_T$ on the specific $\mathcal{H}$ space. Meanwhile, the const parameter $\lambda$ satisfies that

$$\lambda = \epsilon_{D_S}(h^\star) + \epsilon_{D_T}(h^\star), \quad h^\star = \arg\min_{h \in \mathcal{H}} \epsilon_{D_S}(h) + \epsilon_{D_T}(h) \tag{12}$$

where $h^\star$ indicates the best classifier with the lowest error sum $\lambda$ of the source error $\epsilon_{D_S}$ and the target error $\epsilon_{D_T}$ on the $\mathcal{H}$ space. Meanwhile, the space $\mathcal{H}\Delta\mathcal{H}$ satisfies

$$\mathcal{H}\Delta\mathcal{H} = \{\eta : \eta(z^\star) = 1\} \tag{13}$$

where define the variable $z^\star$ as

$$z^\star = \{z : h_1(z) \oplus h_2(z), h_1, h_2 \in \mathcal{H}\} \tag{14}$$

where $\oplus$ indicates the XOR operator.

Therefore, regarding the problem of transferring the source domain policy model $\pi_\theta$ into the target domain, it is equivalent to minimizing variables $\epsilon_{D_S}(h)$ and $d_{\mathcal{H}\Delta\mathcal{H}}$. For minimizing the generalized distance $d_{\mathcal{H}\Delta\mathcal{H}}$, we derive as follows

$$
\begin{aligned}
d_{\mathcal{H}\Delta\mathcal{H}}(\tilde{D}_S, \tilde{D}_T) &= 2 \sup_{h_1, h_2 \in \mathcal{H}} \left| \Pr_{\tilde{D}_S}[\{z : h_1(z) \neq h_2(z)\}] - \Pr_{\tilde{D}_T}[\{z : h_1(z) \neq h_2(z)\}] \right| \\
&= 2 \sup_{\eta \in \mathcal{H}\Delta\mathcal{H}} \left| \Pr_{\tilde{D}_S}[\{z : \eta(z) = 1\}] - \Pr_{\tilde{D}_T}[\{z : \eta(z) = 1\}] \right| \\
&\leq 2 \sup_{\eta \in \mathcal{H}_d} \left| \Pr_{\tilde{D}_S}[z : \eta(z) = 1] - \Pr_{\tilde{D}_T}[z : \eta(z) = 1] \right|
\end{aligned}
\tag{15}
$$

where $\mathcal{H}_\lceil$ denotes the trained classfier space such that $h_1, h_2 \in \mathcal{H}$.

Besides, as the number of target-specific state-action pairs increases, the difficulty of transferring the policy model from the source domain to the target domain increases from a geometric multiple. According to the Generalization Bound theoremZhang et al. (2019), we have

$$
\epsilon_{D_T}(f) < \epsilon_{D_S}^{(\rho)}(f) + d_{f,\mathcal{F}}^{(\rho)}(\tilde{D}_S, \tilde{D}_T) + \lambda + 2\sqrt{\frac{\log\frac{2}{\delta}}{2n}} + \sqrt{\frac{\log\frac{2}{\delta}}{2m}}
$$
$$
+ \frac{2k^2}{\rho}\mathfrak{R}_{n,D_S}(\Pi_1\mathcal{F}) + \frac{2k}{\rho}\mathfrak{R}_{n,D_S}(\Pi_\mathcal{H}\mathcal{F}) + \frac{2k}{\rho}\mathfrak{R}_{m,D_T}(\Pi_\mathcal{H}\mathcal{F}) \tag{16}
$$

where $f$ denotes all scoring functions, $k$ represents the number of categories for classification problems in the source and target domains, $\rho$ is a given const parameter, $\lambda$ is a constant independent of $f$, $\mathfrak{R}$ represents the Rademacher complexity, and $\Pi_1\mathcal{F}$ is defined as

$$
\Pi_1\mathcal{F} = \{x \rightarrow f(x,y)|y \in \mathcal{Y}, f \in \mathcal{F}\} \tag{17}
$$

It can be seen from the above theorem that the increase of the number of categories $k$ will lead to the increase of generalization error in the target domain. In our derivation, more target-specific state-action pairs mean more classification of action categories in both the source and target domains. That is to say, as the difference between the source domain and the target domain becomes larger, the generalization error between the source domain and the target domain will continue to increase. Furthermore, we can get that the increase of target-specific state-action pairs will make it more difficult for the policy model to transfer from the source domain to the target domain environment.

$\square$

**Theorem 2:** Assume $p(s,a)$ is the joint distribution of state $s$ and action $a$. Given $\delta \in [0,1]$, let a partition $\Omega \subseteq \mathbb{R}^n$ on the $\mathcal{H}$ space satisfies $P_{p(s,a)}(s \in \Omega) = \delta$, then
(1) there exists a partition $\Omega_{D_S}$ and $\Omega_{D_T}$ such that

$$
d_{\mathcal{H}\Delta\mathcal{H}}(D_S, D_T) \geq 2\left|E_{s \sim D_S}[A(s) \neq A^{'}(s)] - E_{s \sim D_T}[A(s) \neq A^{'}(s)]\right| \tag{18}
$$

(2) maximizing the MMD loss is equivalent to increasing the distance $d_{\mathcal{H}\Delta\mathcal{H}}$.

**Proof:** First, we prove that there exists a large upper bound of $d_{\mathcal{H}\Delta\mathcal{H}}$ that satisfies the transfer learning constraints for the transfer error $\epsilon_{D_T}$.

Let $p(s,a)$ be the joint distribution of the state $s$ and action $a$. A partition $\Omega \subseteq \mathbb{R}^n$ is constructed such that all states $s$ in this partition $\Omega$ satisfies that

$$
P_{p(s,a)}(s \in \Omega) = \delta \tag{19}
$$

where the variable $\delta \in [0,1]$. Given a classifier $h$, a classification method $k(z) = 1$ is generated on it, where $z \in \{z|h(z) > 0.5\}$. When $\delta = 1$, the partition $\Omega$ uniquely corresponds to a classifier $k$. In this case, the generalization error is

$$
\epsilon(\Omega) = \epsilon(k) = \mathbb{E}(|a - A(s)|) \tag{20}
$$

The optimal partition of the probability distribution $p$ is denoted as

$$
\Omega_p^\star = \underset{\Omega \subseteq \mathbb{R}^n}{\arg\min}\, \epsilon(\Omega) \tag{21}
$$

For the transfer problem on the target domain $D_T$, it is equivalent to the optimization problem

$$
\min_{f,h} \epsilon_{D_S}(h \circ f), \quad \text{s.t.}\ f(s_{D_S}) = f(s_{D_T}) \tag{22}
$$

For the transferred classifier $A^{\mathrm{tran}}$, it belongs to the set of classifiers $\mathbb{A}^\star$ that satisfy

$$
\epsilon_{D_S}(h_{A^{\mathrm{tran}}} \circ f_{A^{\mathrm{tran}}}) \leq \epsilon_{D_S}(\Omega_{D_S}^\star) \tag{23}
$$
$$
f_{A^{\mathrm{tran}}}(s_{D_S}) = f_{A^{\mathrm{tran}}}(s_{D_T}) \tag{24}
$$

Consider the feature function $f_\Omega(s)$ defined as follows: for a given partition $\Omega$,

$$
f_\Omega(s) = \begin{cases} 1_m, & s \in (S_{D_S} \cap \Omega_{D_S}^\star) \vee (S_{D_T} \cap \Omega) \\ 0_m, & \text{otherwise} \end{cases} \tag{25}
$$

where the parameter $m$ denotes the dimensions of the feature vector. Let the classifier be $h(1_m) = 1$. Obviously, $A = h \circ f_\Omega(s) \in \mathbb{A}^\star$. Construct that

$$\hat{\Omega} = \underset{\Omega \subseteq \mathbb{R}^n}{\arg\max} \, \epsilon_{D_T}(\Omega) \quad \text{s.t.} \quad P_{D_T}(s \in \Omega) = P_{D_S}(s \in \Omega^\star_{D_S}) \tag{26}$$

the generalization error of the classifier

$$\hat{A} = h \circ f_{\hat{\Omega}(s)} \in \mathbb{A}^\star \tag{27}$$

corresponding to this partition is

$$\epsilon_{D_T}(\hat{A}) = \max_{A \in \mathbb{A}^\star} \epsilon_{D_T}(A) \tag{28}$$

Define that

$$S^{\text{same}} = \{s | s \in S_{D_S} \cap S_{D_T}\} \quad \text{and} \quad S^{\text{diff}} = \{s | s \notin S_{D_S} \cap S_{D_T}\} \tag{29}$$

Assume that

$$P_{D_T}(s \in \Omega^\star_{D_T}) = P_{D_S}(s \in \Omega^\star_{D_S}) = 0.5 \tag{30}$$

and $s \in S^{\text{diff}} \cap \hat{\Omega}$, this approach still achieves optimization of source domain error while mapping $D_S$ and $D_T$ to the same distribution. In this case, it holds that

$$\max_{A \in \mathbb{A}} \epsilon_{D_T}(A) \geq (1 - \frac{|S^{\text{same}}|}{|S_{D_S} \cup S_{D_T}|})(1 - \epsilon_{D_T}(\Omega^\star_{D_T})) \geq 1 - \epsilon_{D_T}(\Omega^\star_{D_T}) \tag{31}$$

When $S^{\text{same}} = \varnothing$, it degenerates to

$$\max_{A \in \mathbb{A}} \epsilon_{D_T}(A) \geq 1 - \epsilon_{D_T}(\Omega^\star_{D_T}) \tag{32}$$

This implies the existence of worst-case solutions that satisfy the original transfer learning conditions.

A worst-case classifier can be constructed as follows: Let $\Omega_{D_S}$ and $\Omega_{D_T}$ be chosen such that $s$ has an equal probability of occurring in both the source and target domains and $S^{\text{same}} = \varnothing$. Define the feature function

$$f(s) = 1_m \quad \text{if } s \in (S_{D_S} \cap \Omega_{D_S}) \vee (S_{D_T} \cap \Omega_{D_T}) \tag{33}$$

$$f^{'}(s) = 1_m \quad \text{if } s \in (S_{D_S} \cap \Omega_{D_S}) \vee (S_{D_T} \cap (\mathbb{R}^n \backslash \Omega_{D_T})) \tag{34}$$

and let the classifier be $h(1_m) = 1$. For the classifiers $A = h \circ f$ and $A^{'} = h \circ f^{'}$, both belong to classifiers that satisfy the transfer conditions, but there exists a $\mathcal{H}\Delta\mathcal{H}$ lower bound of

$$d_{\mathcal{H}\Delta\mathcal{H}} \geq 2|E_{s \sim D_S}[A(s) \neq A^{'}(s)] - E_{s \sim D_T}[A(s) \neq A^{'}(s)]| = 2 \tag{35}$$

the maximum value of $\mathcal{H}\Delta\mathcal{H}$ is achieved in this case.

Next, we aim to prove that increasing the MMD leads to an increase in the transfer error $\epsilon_{D_T}$. The MMD distance is defined as

$$MMD(X, Y) = \|\frac{1}{n} \sum_i^n \phi(x_i) - \frac{1}{m} \sum_j^m \phi(y_j)\|_H^2$$

$$= \|\frac{1}{n^2} \sum_i^n \sum_{i'}^n \phi(x_i)\phi(x_{i'}) - \frac{2}{nm} \sum_i^n \sum_j^m \phi(x_i)\phi(y_j) + \frac{1}{m^2} \sum_j^m \sum_{j'}^m \phi(y_j)\phi(y_{j'})\|_H$$

$$= \|\frac{1}{n^2} \sum_i^n \sum_{i'}^n k(x_i, x_{i'}) - \frac{2}{nm} \sum_i^n \sum_j^m k(x_i, y_j) + \frac{1}{m^2} \sum_j^m \sum_{j'}^m k(y_j, y_{j'})\|$$

$$= \|\mathbb{E}(k(x_i, x_{i'})) - 2\mathbb{E}(k(x_i, y_j)) + \mathbb{E}(k(y_j, y_{j'}))\|$$

$$\tag{36}$$

where $x_i \sim X$ and $y_j \sim Y$, and the Gaussian kernel function is

$$k(u, v) = e^{-\frac{\|u-v\|^2}{\sigma}} \tag{37}$$

Consider these extreme scenarios:

i) When fixing $\mathbb{E}(k(x_i, x_{i'})) = 1$ and $\mathbb{E}(k(y_j, y_{j'})) = 1$, maximizing the MMD is equivalent to setting $\mathbb{E}(k(x_i, y_j)) = 0$. By using the kernel function $k(u, v)$, we have

$$\mathbb{E}(k(x_i, y_j)) = e^{-\frac{\mathbb{E}(\|x_i - y_j\|^2)}{\sigma}} = 0 \tag{38}$$

which is equivalent to $\mathbb{E}(\|x_i - y_j\|^2) \to +\infty$. Furthermore,

$$\begin{aligned}
\mathbb{E}(\|x_i - y_j\|^2) &= \mathbb{E}(\|x_i\|^2 - 2\|x_i\|\|y_j\| + \|y_j\|^2) \\
&= \mathbb{E}(\|x_i\|^2) - 2\mathbb{E}(\|x_i\|\|y_j\|) + \mathbb{E}(\|y_j\|^2) \\
&= \mathbb{E}^2(\|x_i\|) - 2\mathbb{E}(\|x_i\|)\mathbb{E}(\|y_j\|) - 2\|\operatorname{Cov}(X, Y)\| + \mathbb{E}^2(\|y_j\|) \\
&= (\mathbb{E}(\|x_i\|) - \mathbb{E}(\|y_j\|))^2 - 2\|\operatorname{Cov}(X, Y)\| \\
&\sim +\infty
\end{aligned} \tag{39}$$

This is equivalent to that $\|\overline{x} - \overline{y}\| \to +\infty$.

ii) When fixing $\mathbb{E}(k(x_i, y_j)) = 0$, maximizing the MMD is equivalent to setting $\mathbb{E}(k(x_i, x_{i'})) = 1$ and $\mathbb{E}(k(y_j, y_{j'})) = 1$. As before, this is equivalent to $\mathbb{E}(\|x_i - x_{i'}\|) \to 0$ and $\mathbb{E}(\|y_j - y_{j'}\|) \to 0$. Without loss of generality, we can assume that $\|x_i\| \geq \|x_{i'}\|$ for $x_i, x_{i'} \sim X$. We consider the following on $X$:

$$\begin{aligned}
\mathbb{E}(\|x_i - x_{i'}\|^2) &= \mathbb{E}(\|x_i\|^2 - 2\|x_i\|\|x_{i'}\| + \|x_{i'}\|^2) \\
&= \mathbb{E}_i(\mathbb{E}_{i'}(\|x_i\|^2) - 2\mathbb{E}_{i'}(\|x_i\|\|x_{i'}\|) + \mathbb{E}_{i'}(\|x_{i'}\|^2)) \\
&= \mathbb{E}_i(\|x_i\|^2 - 2\|x_i\|\mathbb{E}_{i'}(\|x_{i'}\|) + \mathbb{E}_{i'}(\|x_{i'}\|^2)) \\
&= \mathbb{E}(\|x_i\|^2 - 2\overline{x}\|x_i\| + \mathbb{E}(\|x\|^2)) \\
&= 2(\mathbb{E}(X^2) - \mathbb{E}^2(X)) \\
&= 2\mathbb{D}(X) \\
&\to 0
\end{aligned} \tag{40}$$

This is equivalent to $\mathbb{D}(X) \to 0$ and $\mathbb{D}(Y) \to 0$.

In summary, when optimizing the MMD, as it approaches the limit, we have

$$\lim_{\text{MMD} \to max} \overline{x} - \overline{y} = +\infty \tag{41}$$

$$\lim_{\text{MMD} \to max} \mathbb{D}(x) = 0 \tag{42}$$

$$\lim_{\text{MMD} \to max} \mathbb{D}(y) = 0 \tag{43}$$

Considering the properties of limits, it is necessary that there exists a real number $\lambda$ such that when $\text{MMD} > \lambda$, $\overline{x} - \overline{y}$ increases monotonically and $\mathbb{D}(x)$ and $\mathbb{D}(y)$ decrease monotonically. This means that there is a critical step after which the MMD training always descends the gradient towards the optimization of $\overline{x} - \overline{y}$, $\mathbb{D}(x)$, and $\mathbb{D}(y)$.

Considering with the Equation (31), when fixing other conditions and only considering the increase of $\overline{f}(s_{D_S}) - \overline{f}(s_{D_T})$, it is equivalent to a decrease in $|S^{\text{same}}|$, which leads to an increase in $\epsilon_{D_T}(A)$.

Consider the feature extraction function $f_\Omega(s) = 1_m$ for a given partition, where $s \in (S_{D_S} \cap \Omega_{D_S}^\star) \vee (S_{D_T} \cap \Omega)$. When fixing other conditions and considering the decrease of $\mathbb{D}(f(s_{D_S}))$ and $\mathbb{D}(f(s_{D_T}))$, we consider the conditions $(S_{D_S} \cap \Omega_{D_T}^\star) \vee (S_{D_T} \cap \Omega_{D_T}^\star)$ and $(S_{D_S} \cap \Omega_{D_S}^\star) \vee (S_{D_T} \cap (\mathbb{R}^n \backslash \Omega_{D_T}^\star))$ To minimize the variance and achieve the optimal partition in the source domain, while ensuring that $|\Omega_{D_T}^\star \cap \Omega_{D_S}^\star|$ approaches $|\mathbb{R}^n \backslash \Omega_{D_T}^\star \cap \mathbb{R}^n \backslash \Omega_{D_S}^\star|$, the positive samples in the source domain and negative samples in the target domain are constrained to a point in the feature space. Similarly, this is also true for the negative samples in the source domain and positive samples in the target domain. Therefore, there exist only the optimal classifiers for $D_S$ and $D_T$ respectively in this feature space, and there does not exist a classifier that is optimal for both domains. Moreover, the partition boundary between the source and target classifiers is orthogonal.

In a word, maximizing the MMD loss is equivalent to increasing the distance $d_{\mathcal{H} \Delta \mathcal{H}}$.

$\square$

# D  IMPLEMENTATION DETAILS

## D.1  NETWORK ARCHITECTURE

To build the Actor and Critic models, we use a three-layer MLP structure on the MuJoCo environment. The first two MLP layers act as feature extractors, while the last MLP layer is used as either the Policy Net or Value Net. The first two MLP layers are followed by a tanh activation function layer. The output of the last MLP layer of the Actor model is the mean value of the output policy, and the output of the last MLP layer of the Critic model is the estimated value of the current state.

## D.2  HYPER PARAMETERS

In the Environment Randomization module, the scale parameter $c$ is set to 1.5 for body mass, body inertia, and geom friction, and 1.3 for dof damping in the MuJoCo environment. For the tunable hyperparameters $\epsilon_1$ and $\epsilon_2$ are set to 0.1, $\epsilon_3$, and $\epsilon_4$, $\epsilon_1$, $\epsilon_2$, and $\epsilon_3$ are set to 0.5 for each experiment, while $\epsilon_4$ is set to 1 for the HalfCheetah-v3 and Hopper-v3 experiments and 3 for the Ant-v3 experiment. The tunable hyperparameter $\tau$ is set to 0.7 for the HalfCheetah-v3 and Ant-v3 experiments, and 0.8 for the Hopper-v3 experiment.

In the Transfer-Controllable Training module, the learning rate of the normal training is set to 3e-4, and the learning rate of the reverse training is set to 3e-5 for each experiment. The total buffer size is set to 4096, with the source domain dataset and the target domain dataset each being 2048, respectively. The step per epoch is set to 30000, and the step per collect is set to 2048. The batch size is set to 64, and the repeat per collect is set to 10. The thread number for collecting data is set to 64 during the model training process.

For the PPO algorithm, we employ both reward normalization and observation normalization techniques. In the loss function, the value function coefficient is set to 0.25, the entropy coefficient is set to 0.0, and the GAE lambda parameter is set to 0.95. Additionally, the epsilon clip parameter is set to 0.2.

## D.3  UNAUTHORIZED TARGET DOMAIN ENVIRONMENTS ON TRAINING PROCESS

In the experiment, we use 32 threads to collect the state-action pair data on the source domain environment, while using 32 threads to collect the corresponding data on the unauthorized target domain environment. In order to ensure the diversity of data collected on the target domain environments, every 4 threads collect the state-action pair data obtained on the target domain environments with the same parameter configuration in parallel. In the subsequent supplementary experimental results section, we demonstrate that using this configuration for can achieve better training performance.

# E  ALGORITHM DETAILS

## E.1  ENVIRONMENT RANDOMIZATION

As indicated in the main text of the paper, the process of Algorithm 1 can be divided into four main phases: fine-tune authorized model, parameter randomization, unauthorized model fine-tuning and screening environment.

In the phase of fine-tuning the authorized model, randomly select one from the source-domain policy models and transfer it to the authorized target domain environment given by the user, as shown in lines 1-2 of Algorithm 1.

In the parameter randomization phase, an unauthorized target domain environment is generated according to some custom randomization rules, as shown in lines 5-6 in Algorithm 1. This phase is mainly to randomize the relevant parameters in the source domain environment according to the characteristics of the source domain environment, so as to obtain the target domain environments similar to the MDP of the source domain environment.

In the unauthorized model fine-tuning phase, an Actor model $\pi_\theta$ is randomly selected from the source domain model set $P_{model}$ for transfer learning, as shown in lines 7-8 in Algorithm 1. Then,

we retrain the Actor model initialized by $\pi_\theta$ in the generated target domain environments. In the MuJoCo environment used in the verification of this paper, this simple fine-tune method has been able to verify the availability of our framework. In practice, the appropriate transfer learning algorithm could be selected according to the environment characteristics.

In the screening environment phase, the unauthorized target domain environments are selected through the given custom rules, as shown in lines 10-11 in Algorithm 1. In this paper, we use the converge time and the source domain rewards to judge whether a target domain environment is easy to transfer. It is a simple and effective method to select the suitable target domain environments.

### E.2    TRANSFER-CONTROLLABLE TRAINING

As indicated in the main text of the paper, the data processing flow of Algorithm 2 includes four main phases: algorithm preparation, data collection, auxiliary variable calculation, and model parameter update.

In the preparation phase, it mainly completes the construction of the environments and the initialization of related variables, as shown in lines 1-3 in Algorithm 2. The initial parameters of the target domain environments come from the result of Algorithm 1.

In the data collection phase, the Actor model $\pi_{\theta_k}$ is used to collect trajectories on the source domain environment $E$ and the target domain environments $\{E_k\}_{k=1}^L$ respectively, as shown in lines 5-10 in Algorithm 2. When the sum of the capacities of the data buffer $\mathcal{D}_{\text{Source}}$ and $\mathcal{D}_{\text{Unauth\_Target}}$ is greater than the maximum threshold $|\mathcal{D}|$, the data collection phase ends.

In the auxiliary variable calculation phase, the discounted reward $\hat{R}_t$ and the advantage estimates $\hat{A}_t$ required for the subsequent training phase are calculated, as shown in lines 11-12 in Algorithm 2. Calculate $\hat{A}_t^{\mathcal{P}}$ using the generalized advantage estimation method on the data buffers $\mathcal{D}_{\text{Source}}$ and $\mathcal{D}_{\text{Unauth\_Target}}$.

In the model parameter update phase, the four loss functions, shown in Fig. 3, are used to update the model parameters, and the specific process is shown in lines 15-23 in Algorithm 2. In lines 14-15, the preparations before model training is completed. Then the MMD loss $L_{\text{MMD}}$ and the Actor loss $J_{\text{TCRL}}^{\theta_k}(\theta)$ are calculated through Eq. (4) and Eq. (3), respectively. Next, the Critic loss is calculated through Eq. (9). Finally, the model parameters of the Actor network $\pi_{\theta_k}$ and the Critic network $v_{\phi_k}$ are updated using the gradient ascent method, as shown in lines 16-18.

## F    SUPPLEMENTARY EXPERIMENTAL RESULTS

### F.1    ABLATION STUDIES ON EACH COMPONENT

To comprehensively evaluate the contribution of each key component within our proposed TCRL framework, we conducted additional ablation studies on the HalfCheetah-v3 benchmark. The results, presented in Table 3, quantify the impact of environment randomization and the transfer-controllable training module (specifically, the MMD loss).

Table 3: Ablation Studies on Environment Randomization and Transfer-Controllable Training Module. **w/o Env Filtering** refers to the variant where the process of screening and excluding unauthorized target environments during training is removed. **w/o MMD** indicates the removal of the Maximum Mean Discrepancy (MMD) loss from the policy model's objective function.

| Reward | w/o Env Filtering | w/o MMD | TCRL (full) |
|---|---|---|---|
| Unauthorized | 1918 | 3012 | 2516 |
| Authorized | 3098 | 3985 | 4207 |

The experimental results highlight the significance of both components:

**Impact of Environment Filtering:** When Environment Filtering is omitted ("w/o Env Filtering"), the model's performance degrades substantially. The reward in authorized scenarios drops from 4207 (TCRL full) to 3098. Concurrently, the reward in unauthorized scenarios is the lowest at

1918. This outcome aligns with our hypothesis: without environment filtering, the training process is exposed to unauthorized domains that may include dissimilar target environments. Such exposure negatively impacts the model's ability to learn an effective policy for authorized tasks and to generalize appropriately.

**Impact of MMD Loss:** Removing the MMD loss ("w/o MMD") while retaining environment filtering also leads to a noticeable performance decline compared to the full TCRL model. The authorized reward decreases to 3985 from 4207, and the unauthorized reward increases to 3012 from 2516. The MMD loss is designed to encourage the policy to learn domain-invariant representations of state-action pairs, thereby helping to distinguish and adapt behaviors between authorized and unauthorized domains. Without it, the model struggles to effectively capture these crucial state-action differences, leading to suboptimal performance in authorized settings and increased undesirable behavior in unauthorized ones.

In contrast, the TCRL (full) model, which integrates both Environment Filtering and the MMD loss, achieves the highest reward (4207) in authorized environments while maintaining a comparatively lower reward (2516) in unauthorized environments. This demonstrates the synergistic effect of these components in enabling robust and controllable transfer learning.

## F.2 DIFFERENT TARGET DOMAIN ENVIRONMENT CONFIGURATIONS ON TCRL TRAINING

In this experiment, we mainly verify the impact of different unauthorized target domain environment number configurations on model performance during the training process. It mainly includes changes in the total number of authorized target domain environments and changes in the proportion of environments with the same configuration in the total target domain environments. In all these experiments, we use 32 threads to collect data from the source domain.

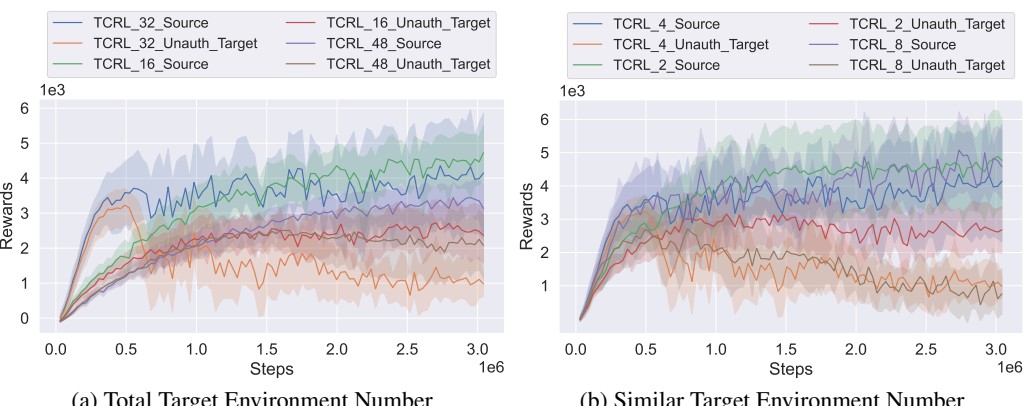

(a) Total Target Environment Number      (b) Similar Target Environment Number

Figure 8: Training performance of different unauthorized target domain environment configurations on the HalfCheetah-v3 environment. The variables $TCRL\_x\_Source$ and $TCRL\_x\_Unauth\_Target$ represent the training performance of the TCRL algorithm on the source and unauthorized target domains, respectively. (a) In this experiment, $x$ refers to the number of threads utilized for data collection in the target domain environments. Specifically, every 4 threads were assigned to use the unauthorized target domain environments having an identical configuration. (b) In this experiment, $x$ denotes that every group of $x$ threads was allocated to collect data from a target domain environment having the identical configuration. In all these experiments, we use 32 threads to collect data on the source domain.

In the Total Target Environment Number experiment, we change the total number of target domain environments, as shown in the left part of Fig. 8. The TCRL_32_Source curve and the TCRL_32_Unauth_Target curve represent the default TCRL algorithm training configuration. Comparing the TCRL_32_Unauth_Target curve with the TCRL_16_Unauth_Target curve, it can be seen that reducing the total number of target domain environments will increase the reward value achieved by TCRL in the target domain, which also means that reducing the total number of target domain environments reduces the effectiveness of TCRL algorithm in suppressing target domain performance.

Comparing the TCRL_32_Source curve with the TCRL_48_Source curve, it can be seen that increasing the total number of target domain environments will reduce the reward value obtained by TCRL in the source domain. This means that increasing the total number of target domain environments will reduce TCRL's performance in the source domain. Overall, the default configuration of the total number of target domain environments is a more suitable training parameter configuration.

In the Similar Target Environment Number experiment, we change the proportion of environments with identical configuration in the total target domain environments, as shown in the right part of Fig. 8. TheTCRL_4_Source curve and the TCRL_4_Unauth_Target curve represent the default training configuration of the TCRL algorithm. In the source domain, different configurations achieve similar reward values, with similar trends for the TCRL_4_Source curve, the TCRL_2_Source curve and the TCRL_8_Source curve. In the target domain, the reward value obtained by the TCRL_2_Unauth_Target curve is higher than that of the other two dotted lines, which means that this configuration weakens the performance suppression effect of TCRL on the target domain. That is to say, the training performance is poor when the target domain environment where data is collected by every two threads is the same. Overall, the default configuration of the identical target environment proportion in the total target domain environments is a more suitable training parameter configuration.

### F.3  REWARD SCALE FOR TARGET DOMAIN ENVIRONMENTS

In this experiment, we mainly aim to verify the impact of target domain environments with different random parameters on the final reward value obtained by the policy model. As shown in Fig. 9, the final reward value obtained by the policy model in the randomized target domain environment using the same randomization control parameters has a significant variance. To ensure the objectivity of the experimental results, we scale the reward values based on the final reward values obtained in the source and target domains.

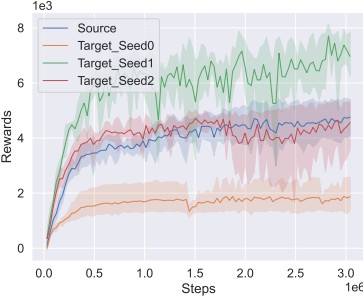

Figure 9: The impact of target domain environment with different randomization parameters on the final reward value. The $Soure$ line represents the reward curve on the source domain environment, and the other three lines $Target\_seedx$ indicate the reward curves on the different randomized target domain environments.

### F.4  HYPERPARAMETER SENSITIVITY

We conducted experiments on the HalfCheetah-v3 environment to analyze the sensitivity of key hyperparameters in our framework:

$\delta$ **for environment perturbation.** Since $\delta$ is determined through iterative optimization as described in our Q2 response, we tested how reducing this parameter affects protection capability:

Table 4: Sensitivity to environment perturbation $\delta$. "orig." refers to the original reward.

| Perf. | $\delta$ (orig.) | $\delta/2$ | $\delta/4$ | PPO_Trans |
|---|---|---|---|---|
| HalfCheetah-v3 | 2516 | 2558 | 2617 | 4115 |

TCRL maintains effective protection even with reduced perturbation amplitude.

$T_{threshold}$ **for quick transfer.** This parameter defines the time threshold for identifying quickly transferable environments (20% of training time from scratch). Testing variations of this threshold shows:

Table 5: Sensitivity to quick transfer threshold $T_{threshold}$.

| Perf. | $0.8T_{threshold}$ | $T_{threshold}$ | $1.2T_{threshold}$ |
|---|---|---|---|
| Unauth | 2461 | 2516 | 2671 |
| Auth | 4195 | 4207 | 4015 |

The parameter exhibits moderate sensitivity without substantially impacting protection.

**MMD loss weight $\eta$.** This parameter balances feature distribution separation between domains. Testing values around our default (3e-5):

Table 6: Sensitivity to MMD loss weight $\eta$.

| Perf. | 1e-5 | 3e-5 | 5e-5 |
|---|---|---|---|
| Unauth | 2608 | 2516 | 2497 |
| Auth | 4224 | 4207 | 4195 |

Results show low sensitivity within this range.

**KL divergence weight $\lambda$.** This parameter controls the influence of authorized policy behavior:

Table 7: Sensitivity to KL divergence weight $\lambda$.

| Perf. | 1e-2 | 1e-3 | 1e-4 |
|---|---|---|---|
| Unauth | 2647 | 2516 | 2623 |
| Auth | 3872 | 4207 | 4007 |

$\lambda$ shows higher sensitivity than other parameters. Our default value (1e-3) provides optimal balance between maintaining authorized performance while limiting unauthorized performance.

Most parameters show low to moderate sensitivity, with $\lambda$ requiring the most careful tuning.

F.5 EXPERIMENTAL RESULTS OF OTHER MUJOCO ENVIRONMENTS

The experimental results of the other three MuJoCo environments, such as InvertedDoublePendulum, Walker2d and Humanoid, as shown in Fiugre 10 and Fig. 11.

In Fig. 10, the baseline PPO algorithm and our TCRL algorithm can achieve similar rewards in the source domain. During the training process, the rewards obtained in the target domain are much less than the rewards in the source domain. In particular, in the Humanoid environment, the green reward curve of the target domain has basically no upward trend.

In Fig. 11, the orange reward curves initialized by our TCRL model achieve the worst results, which means that the TCRL model can prevent the migration of the policy model from the source domain to the unauthorized target domain to a certain extent. Meanwhile, the blue reward curves intialized by the original PPO model can obtain similar results with the green reward curves of random intialization. It means that the original PPO model cannot prevent the source domain policy models transfer to the target domain.

In general, these experimental results are similar to those of the three mujoco environment experiments in the main text, which can support the relevant statements in the main text.

F.6 EXPERIMENTAL RESULTS ON HAND MANIPULATION SUITE ENVIRONMENT

In this experiment, we are examining the impact of two transfer reinforcement learning algorithms, namely the DAPG algorithmRajeswaran et al. (2017a) and the REvolveR algorithmLiu et al. (2022),

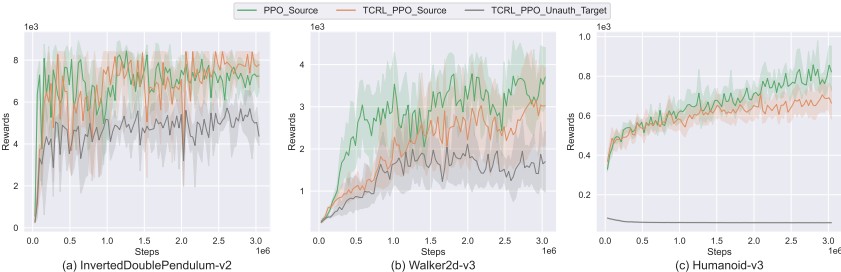

Figure 10: Training performance of the baseline PPO algorithm and our TCRL algorithm. The blue $PPO\_Source$ and orange $TCRL\_PPO\_Source$ solid curves denote the performance of PPO and TCRL on the source domain, while the green $TCRL\_PPO\_Unauth\_Target$ dotted curves indicate the performance of TCRL on the target domains.

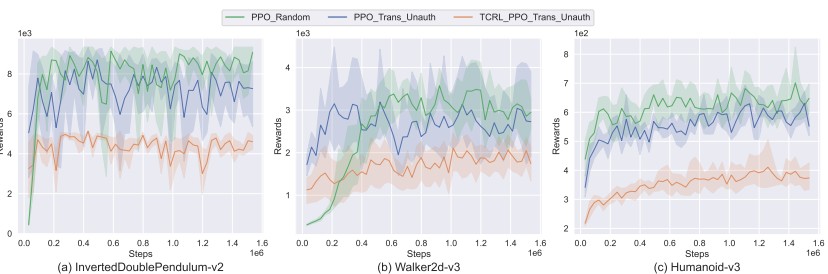

Figure 11: Comparing the transfer performance of the PPO and TCRL models on the target domain, the blue $PPO\_Trans\_Unauth$ curve denotes the PPO model and the orange $TCRL\_PPO\_Trans\_Unauth$ curve denotes the TCRL model. The blue $PPO\_Random$ curve, trained with a random initialized model, serves as the baseline.

on the transfer-controllability of the TCRL model. The objective is to evaluate the effectiveness of these algorithms in attacking the transfer-controllability of the TCRL model. In these experiments, we replaced the PPO algorithm in the main text with the NPG algorithm.

### F.6.1 HAND MANIPULATION SUITE ENVIRONMENT

This part of the experiment is carried out on the hand manipulation suite environmentLiu et al. (2022). This environment is constructed based on the ADROIT platformRajeswaran et al. (2017a), as shown in Fig. 12.

In Fig. 12, the ADROIT platform is a 24-DoF anthropomorphic platform designed for addressing challenges in dynamic and dexterous manipulation. The first, middle, and ring fingers have 4 degrees of freedom (DoF). Little finger and thumb have 5 DoF, while the wrist has 2 DoF. Each DoF is actuated using position control and is equipped with a joint angle sensor. In this experiment, we use two kinds of these tasks, the object relocation task and the door opening task. As shwon in Fig. 12 (a), the goal of the object relocation task is to move the blue ball to the green target. As shwon in Fig. 12 (b), the goal of the door opening task is to undo the latch and swing the door open.

The hand manipulation suite environmentLiu et al. (2022) is designed to make some evolving transferable environments for transfer reinforcement learning, as shown in Fig. 13. The evolutionary generation process of the transferable five-finger dexterous hand robot is shown in Fig. 13 (c). In the beginning, the hand robot had five dexterous fingers. In the process of continuous evolution, the middle finger, ring finger, and little finger of the robot are getting shorter and shorter. In the end, the hand robot only retained two fingers such as the thumb and index finger, and only had 1 DoF.

Next, we can construct the transferable learning tasks as shown in Fig. 13 (a) and (b). In the object relocation transfer task, the objective of the source domain task is to move a blue ball to the green target using the original five-finger dexterous hand robot. However, in this case, the robot is substi-

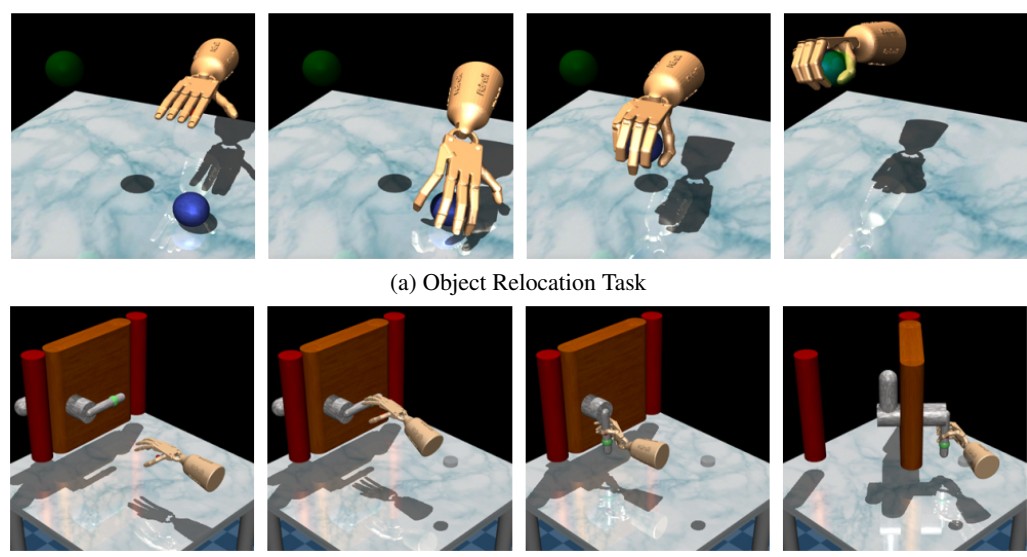

(a) Object Relocation Task

(b) Door Opening Task

Figure 12: The five-finger dexterous hand provided in the ADROIT platformRajeswaran et al. (2017a).

tuted with a simpler two-finger robot in the target domain. Similarly, in the door opening transfer task, the robot configuration remains unchanged, but the objective is modified to opening the door.

### F.6.2 EXPERIMENTAL RESULTS AND ANALYSIS

The experimental results are shown in Table 8 and Table 9 below. In these tables, "From Scratch" means training the policy model from scratch on the target domain, while "Direct Finetune" means using a pre-trained policy model from the source domain for transfer learning. There are two kinds of pre-trained policy model, the "NPG" modelRajeswaran et al. (2017b) and our "TCRL" model. Then, two kinds of transfer reinforcement learning algorithms, the "DAPG" algorithmRajeswaran et al. (2017a) and the "REvolveR" algorithmLiu et al. (2022), are applied to attack the transfer-controllability of the TCRL model. In the "Sparse Reward" setting, only task completion is rewarded. In the "Dense Reward" setting, a distance reward is provided at every step.

In the REvolveR algorithmLiu et al. (2022) and the DAPG algorithmRajeswaran et al. (2017a), an adaptive training scheduling strategy is employed to enhance training efficiency. Consequently, it is not possible to predefine the total number of RL iterations in order to compare performance fairly under the same number of iterations. Instead, the REvolveR algorithmLiu et al. (2022) compares the number of RL optimization steps required to achieve a 90% success rate on the tasks. In this paper, we continue to use the above evaluation method.

From Table 8, none of the transfer learning algorithms initialized with the TCRL model could converge within 100K iterations. The reason may be that the five-finger robot and the two-finger robot grab the blue ball in completely different ways, as shown in Fig. 13 (a). In the TCRL model, due to the reverse training on positive samples in the evolutionary training process, it becomes challenging for transfer reinforcement learning algorithms to obtain positive samples of grasping the blue ball in the target domain. This significantly amplifies the training difficulty for the two-finger robot in the target domain. As a result, the training speed of transfer reinforcement learning using the TCRL model as the initialization model is significantly slowed down in the object relocation task. In other words, the TCRL model has hindered the transfer progress of the DAPG algorithm and the REvolveR algorithm.

From Table 9, the convergence speed of the transfer learning algorithm initialized with the TCRL model is significantly reduced. Compared with the object relocation task, in the door opening task, the execution process of pushing the door handle is similar for the five-fingered robot and the two-

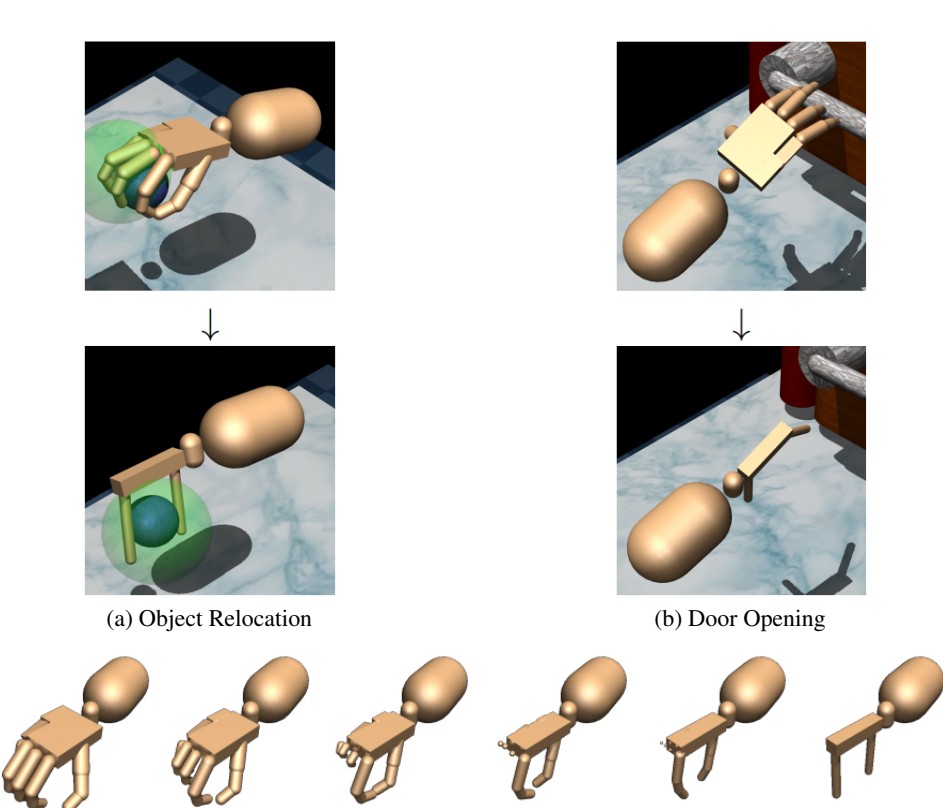

(a) Object Relocation        (b) Door Opening

(c) Transferable Robot Environments

Figure 13: The transferable tasks on hand manipulation suiteLiu et al. (2022).

Table 8: The experimental results of the target transfer task

|  | Dense Reward | | Sparse Reward | |
| --- | --- | --- | --- | --- |
| From Scratch | >100K | | ∞ | |
| Initialized Model | NPG | TCRL | NPG | TCRL |
| Direct Finetune | 43.5K | >100K | ∞ | - |
| DAPGRajeswaran et al. (2017a) | 23.3K | >100K | ∞ | - |
| REvolveRLiu et al. (2022) | - | >100K | 18.1K | >100K |

fingered robot, as shown in Fig. 13 (b). Therefore, in this task, even with the initialization of the TCRL model, the REvolveE algorithmLiu et al. (2022) can still achieve the goal of a success rate exceeding 90%. However, our TCRL model can still significantly slow down the convergence speed of the REvolveE algorithm, which can still generate certain value in practical applications.

Table 9: The experimental results of the door opening transfer task

|  | Dense Reward | | Sparse Reward | |
|---|---|---|---|---|
| From Scratch | - | | $\infty$ | |
| Initialized Model | NPG | TCRL | NPG | TCRL |
| Direct Finetune | 7.6K | 82.5K | $\infty$ | - |
| DAPGRajeswaran et al. (2017a) | 5.4K | 48.3K | $\infty$ | - |
| REvolveRLiu et al. (2022) | - | 45.4K | 2.6K | 58.7K |

Overall, the above experimental results demonstrate that the TCRL model provides a certain level of protection for the intellectual property of the policy model when facing attacks from certain transfer reinforcement learning algorithms.

### F.7 COMPARISON WITH DOMAIN RANDOMIZATION

While traditional domain randomization (e.g., MAML) aims to enhance generalization, TCRL selectively restricts transfer to unauthorized domains. Our supplementary experiments demonstrate TCRL's superior performance:

Table 10: Performance comparison of MAML and TCRL across different domains and environments. Values represent rewards.

| Method | Domain | HalfCheetah-v3 | Hopper-v3 | Ant-v3 |
|---|---|---|---|---|
| MAML | Unauthorized | 2916 | 1475 | 1387 |
| | Authorized | 3972 | 1837 | 1678 |
| TCRL | Unauthorized | 2516 | 1028 | 1043 |
| | Authorized | 4207 | 2075 | 2427 |

These results confirm that directly applying domain randomization techniques to our task would lead to suboptimal outcomes. Our approach with MMD loss and KL divergence constraints achieves the desired balance: limiting performance in unauthorized domains while maintaining or improving it in authorized ones.

## G  DISCUSSION

**Question1:** To protect the policy model, it is advisable to conceal the model parameters and strictly restrict access to an API interface specifically designed for querying policy decisions based on the observed state. Given this approach, is it still necessary to implement a transfer-controllable policy?

**Answer:** Yes, it is still necessary. Suppose Company A has designed a robot $R_A$ and trained the corresponding baseline policy model $\pi_A$. At the same time, Company B has replicated a robot $R_B$ with similar dynamic characteristics and obtained the API of Company A's robot's policy model $\pi_A$. In this case, Company B can use the API to collect the motion trajectories $T_r$ of robot $R_B$ and then use relevant methods of offline reinforcement learning to obtain an approximate version of the policy model $\hat{\pi}_A$. By applying transfer learning to the $\hat{\pi}_A$ model, Company B can obtain a suitable policy model $\pi_B$ for robot $R_B$.

However, when Company A trains the baseline policy model $\pi_A$ using the TCRL algorithm, if Company B tries to use the same API, they would only collect poor-quality motion trajectories. As a result, subsequent offline reinforcement learning and transfer learning processes cannot be carried

out. Therefore, training a transfer-controllable policy model becomes necessary in order to mitigate this issue.

**Question2:** In the paper, the unauthorized target domain environments are designed by randomizing some parameters in the environments. However, it would be quite rare that the real target application is only a few parameters different from the source environments while all other settings are the same.

**Answer:** Yes, perhaps such cases are quite rare. However, if Company B intends to steal the intellectual property of Company A's policy model, they would need to take certain steps to construct a series of similar target domain environments. For example, as shown in Fig. 13 (c), Company B can create a series of intermediate robots that allow Company A's five-finger hand robot to transition naturally to Company B's two-finger hand robot. In general, by using transfer learning algorithms, Company B can avoid some of the errors that Company A would encounter when training from scratch.

**Question3:** The environment randomization module can be time-consuming and may not be suitable for all scenarios.

**Answer:** No single method can be universally applicable to all scenarios, and the environment randomization module is merely a simple preliminary solution. This paper aims to raise awareness about the issue of protecting policy model intellectual property and propose a general solution. In practical applications, various more efficient environment randomization schemes can be designed for this module.

