# OpenReview forum: "Transfer-Controllable Policy for Model Protection in Deep Reinforcement Learning"
_ICLR.cc/2026/Conference — ICLR 2026 Conference Withdrawn Submission_

### Official Review · Reviewer_aFxz · 2025-10-30

**Soundness:** 3
**Presentation:** 2
**Contribution:** 2
**Rating:** 4
**Confidence:** 4

**Summary:**

This paper introduces transfer-controllable RL, a framework for training selectively transferable deep reinforcement learning policies to protect intellectual property by performing effectively in authorized target domains while simultaneously failing when transferred to unauthorized target domains. The framework consists of two main components: an Environment Randomization module that generates unauthorized target domains by perturbing the source environment's parameters, and a transfer-controllable training module. The core of the training module is a composite objective function that combines a standard RL loss for the source domain, a reverse-optimization loss for unauthorized domains, a maximum mean discrepancy loss to separate the feature representations of source and unauthorized states, and a Kullback-Leibler divergence term to maintain performance on authorized domains. The authors validate their approach empirically in both discrete and continuous control environments, demonstrating that their method can create a significant performance gap between authorized and unauthorized policy transfers.

**Strengths:**

- The paper introduces the concept of "transfer-controllable learning" for DRL, which is a valuable contribution. This work shifts the paradigm of DRL IP protection from passive, post-hoc verification methods like watermarking to an active, behavior-shaping defense mechanism embedded directly into the policy's training process. This is an interesting and timely direction for the DRL security community.

- The proposed TCRL framework is systematically described. The two-module design is intuitive. In particular, the authors spell out their composite objective (Eq.2) and provide pseudocode for both the randomization and training steps. The use of an adversarial reward term for unauthorized domains, together with an MMD loss to separate features, is a reasonable engineering approach to achieve the stated goal.

- The experimental results provide a good demonstration of the framework's primary claim. The contrast in transfer performance between authorized and unauthorized target domains (Figure 7, Table 1) effectively validates that the proposed training objective can successfully produce a policy with the desired controllable transferability.

**Weaknesses:**

- The comparison between the proposed method and the SL methods is not comprehensive. The contribution and challenge of using controllable transfer on DRL are not clear.
- The threat model is narrow and lacks convincing evidence. The "unauthorized" transfer is limited to fine-tuning a policy on an environment with slightly randomized physical parameters. However, a primary threat in this domain is not replicating exact model weights but model extraction(or policy stealing) via imitation learning, which the proposed framework offers no defense against.
- In a real-world setting, new but legitimate use cases and deployment environments may emerge even after the initial training is completed. The framework requires the model owner to provide an exhaustive, pre-defined list of all authorized target domains at training time, while relying on random sampling around the source domain to generate unauthorized ones. This might make the result policies inherently brittle because it is trained to fail in any new environment that was not explicitly included in the authorized set, which contradicts the stated goal of “maintaining transferability for future scalability.”
- The authors admit that TCRL "consumes approximately twice the computational resources" of standard training. This is a significant practical limitation that should be a core part of the experimental analysis, not a footnote. The evaluation lacks critical metrics such as total training time, sample complexity, or wall-clock time to convergence, making it impossible to assess the practical trade-offs of the framework.
- In experiments, only DQN and PPO are considered; more DRL algorithms and models should also be considered to evaluate the generalization of the proposed method.
- A minor weakness is presentation: The figures are too small and really difficult to read, and the paper in general contains some writing mistakes, like missing punctuation.

**Questions:**

- Could you provide a detailed empirical analysis of this overhead, including metrics like total samples, wall-clock training time, and GPU hours, comparing TCRL directly against a standard training setup?
- How does TCRL defend against model extraction attacks, where an adversary with only black-box query access can use imitation learning on expert trajectories from the authorized domain to train a high-fidelity, unprotected clone of your policy?
- How is a user meant to define the authorized domain set in practice to ensure future scalability to new, unforeseen but legitimate deployment environments (e.g., a new robot model for a new customer)? Does the framework's inherent brittleness to novel-but-authorized domains not severely undermine its practical utility and scalability?

---

### Official Review · Reviewer_Vh7z · 2025-10-31

**Soundness:** 2
**Presentation:** 1
**Contribution:** 3
**Rating:** 4
**Confidence:** 3

**Summary:**

This paper tackles the problem of learning transfer-controllable policies in DRL. The authors point out that policies trained at significant cost may be illegally repurposed through fine-tuning to unsafe domains due to their transferability. To address this issue, they introduces TCRL, a training framework intended to protect DRL policy IP by making policies easy to transfer in pre-defined authorized target domains while difficult to transfer in unauthorized domains. TCRL comprises two modules: (i) Environment Randomization, which randomly generates and screens candidate unauthorized domains, and (ii) Transfer-Controllable Training, which trains the policy to resist adaptation in those unauthorized candidates while preserving transferability to authorized domains. Experiments in both a maze environment and MuJoCo demonstrate the effectiveness of the approach.

**Strengths:**

The paper addresses a novel and important topic, transfer control in DRL. While similar ideas have been explored for classifiers and LLMs, systematic treatment in DRL has been limited. This work highlights an impactful direction for DRL security and IP protection.
The proposed method is simple and broadly applicable to both on-policy and off-policy algorithms, and comprehensive ablations empirically validate the contribution of each component.

**Weaknesses:**

Scalability to large environments and computational cost: the effectiveness of the method remains untested in large, high-dimensional environments. The Environment Randomization module requires fine-tuning on each candidate, so both the per-candidate cost and the size of the environment-parameter search space grow with environment complexity, making comprehensive coverage of unauthorized domains harder. Consequently, the overall performance may weaken at scale.

Lack of analysis regarding the training settings that an attacker might use for transfer learning: A malicious user can apply arbitrary optimization settings to a pretrained model. For example, the search scope of the Environment Randomization module is restricted to the neighborhood of the source domain, excluding environments that deviate substantially. Consequently, by choosing a large learning rate, an attacker may enable transfer to environments not considered by the Environment Randomization module. Therefore, an analysis of the attacker’s training settings is necessary.

Minor comments:
- The paper contains many errors and is difficult to read. For example, the reference of appendix sections appear to be misnumbered. In addition, the indicator $\mathbb{1}_a$ in Equation (5) does not seem to be defined. Although the appendix provides Symbol Definitions, they are not referenced from the main text, and the notation in the main text are insufficient.
- The screening procedure in the Environment Randomization module is unclear. Algorithm 1 appears to consider a converge time and a scaling factor. However, the paper does not specify how the converge time is defined, nor how these two metrics are used for screening.
- Missing optimization details for the attacker in transfer experiments: ****while training-time hyperparameters for the Transfer-Controllable Training module are specified, the transfer-phase ****optimization settings on unauthorized domains are not reported. A sound assessment requires sensitivity to these attacker hyperparameters; for instance, an overly small learning rate would understate the attainable gains on unauthorized targets and overestimate the method’s protection.

**Questions:**

- How do the fine-tuning hyperparameters used in the Environment Randomization module affect the effectiveness of the proposed method? Intuitively, a smaller fine-tuning budget is likely to reduce the method’s effectiveness because it reduces the accuracy of estimating and screening candidate unauthorized domains.
- In the transfer experiments, what training settings were used for learning in the unauthorized domains?
- How do the attacker’s training settings for transfer affect the effectiveness of the proposed method?

---

### Official Review · Reviewer_bSBq · 2025-11-01

**Soundness:** 2
**Presentation:** 2
**Contribution:** 3
**Rating:** 2
**Confidence:** 3

**Summary:**

This paper introduces a new problem setting in reinforcement learning, called transfer-controllable policy learning, which aims to protect the intellectual property of pretrained policy models by allowing them to be transferable only in authorized environments while preventing transferability to unauthorized environments. To achieve this, the authors propose a framework that synthesizes unauthorized environments and trains the policy across both authorized and unauthorized settings. During policy learning, they introduce new objective functions that encourage strong performance in authorized environments while discouraging success in unauthorized ones. The framework is validated in grid-based environments and three continuous control tasks.

**Strengths:**

- The paper addresses a highly meaningful scenario that could have a positive impact in practical applications.

- The proposed method appears effective in achieving its target: when the pretrained policy is fine-tuned in authorized domains, it maintains strong performance, whereas in unauthorized domains, its performance degrades significantly, demonstrating controlled transferability.

**Weaknesses:**

- There are issues with the problem formulation. It is unclear how the source, authorized, and unauthorized domains are defined. Theorem 1 is introduced, presumably to characterize the separation between authorized and unauthorized domains, but it is poorly justified and lacks interpretation on how it is used within the framework.

- In L234–L235, the authors state that “this module randomly generates target-domain environments and selects those that are easily transferable.” If such “easily transferable” environments are instead considered unauthorized, would TCRL remain effective under this scenario?

- The presentation is difficult to follow and interpret, with several unclear or inconsistent points:
	- L53: "During policy training … making them more vulnerable to theft than large datasets". However, in Figure 1, the red and blue arrows originate from data, not the model—is this intentional?
	- L59–L61: "Competitors may misuse obtained policy models by transferring them to similar scenarios, violating IP rights". Are these similar domains considered authorized or unauthorized? The example in Figure 1 does not clearly illustrate this distinction.
	- Several notations are unclear: In Equation 3, what do $n_1$ and $n_2$ represent? In Equation 4, what is the meaning of $A(s)$ and $A'(s)$?.
	- What is the role of Equation 5 in overall method?
	- In Equation 2, for third term, where is $s_t$ sampled from?
	- Figure 5 lacks captions for panels (a) and (b).

**Questions:**

- Why do you need the set of actor models in Algorithm 1?
- In Figure 7, on which environment (authorized or unauthorized) is PPO_random evaluated?
- In Figure 3, why does the authorized target domain bypass the Feature Net?
- Is the source policy fine-tuned in Algorithm 2? The description is unclear, since in Line 2 of Algorithm 2, the networks are randomly initialized.

---

### Official Review · Reviewer_XdFx · 2025-11-01

**Soundness:** 3
**Presentation:** 1
**Contribution:** 3
**Rating:** 4
**Confidence:** 2

**Summary:**

In this paper, the authors propose Transfer-Controllable Reinforcement Learning (TCRL), to prevent the intellectual property of DRL models. The goal is to make the policies transferable to authorized environments but non-transferable to unauthorized ones. TRCL has two components: Environment randomization generates unauthorized target domains, while Transfer-Controllable Training trains a policy on both source and target domains, and performs reverse optimization on unauthorized ones. Experiments on Maze and MuJoCo environments with DQN and PPO show that TCRL-trained policies retain performance in authorized domains but only approximately 50\% in unauthorized ones.

Even if the idea is novel and aims to solve an interesting problem, the lack of clarity in the paper, small grammatical errors, and the wrong in-text citations decrease the value of the paper.

**Strengths:**

S1. Novelty: A similar concept (non-transferable learning, Wang et al. 2022) was proposed for image classification models, but the paper is the first to explore allowed transferability in DRL applications.

S2 Theoretical Support: Theoretical evidence and mathematical proofs for domain adaptation theory is a good choice.

**Weaknesses:**

W1. Clarity and Presentation Issues: I had a very hard time following the framework. The interaction between two modules, how unauthorized environments are generated and used, is not clearly explained. The algorithms given in the paper do not improve the clarity either.

W2. Conceptual Clarity Issue: From the definition of source-similar domains, I understand that it learns a representation shift instead of transferable domains. Again, in Figure 4, the unauthorized target domain is still the same task, so transfer-controllable training is simply less effective in different dynamics. The terms of authorized and unauthorized domains remain vague.

W3. Conflict with the core claim: Since TCRL trains with data from unauthorized domains, does this conflict with the original premise? The model learns from those domains to resist them, which still requires access.

**Questions:**

Q1. In-text references are given in wrong format. Even in some cases, the year was missing. There are also small grammar errors. The authors should correct every in-text reference and do a grammar check.

**Details Of Ethics Concerns:**

No ethics concerns

---

### Note · Authors · 2025-12-05

I have read and agree with the venue's withdrawal policy on behalf of myself and my co-authors.